# SonicMaster: Towards Controllable All-in-One Music Restoration and Mastering

**Jan Melechovsky** [1]   **Ambuj Mehrish** [2]   **Abhinaba Roy** [1]   **Dorien Herremans** [1]

## Abstract

Music recordings often suffer from audio quality issues such as excessive reverberation, distortion, clipping, tonal imbalances, and a narrowed stereo image, especially when created in non-professional settings without specialized equipment or expertise. These problems are typically corrected using separate specialized tools and manual adjustments. In this paper, we introduce *SonicMaster*, the first unified generative model for music restoration and mastering that addresses a broad spectrum of audio artifacts with text-based control. *SonicMaster* is conditioned on natural language instructions to apply targeted enhancements, or can operate in an automatic mode for general restoration. To train this model, we construct the *SonicMaster* dataset, a large dataset of paired degraded and high-quality tracks by simulating common degradation types with nineteen degradation functions belonging to five enhancements groups: equalization, dynamics, reverb, amplitude, and stereo. Our approach leverages a flow-matching generative training paradigm to learn an audio transformation that maps degraded inputs to their cleaned, mastered versions guided by text prompts. Objective audio quality metrics demonstrate that *SonicMaster* significantly improves sound quality across all artifact categories. Furthermore, subjective listening tests confirm that listeners prefer *SonicMaster*'s enhanced outputs over other baselines. The model and demo samples are available through https://github.com/AMAAI-Lab/SonicMaster.

## 1. Introduction

Music recordings produced in amateur settings often suffer from various quality issues that distinguish them from professionally mastered recordings (Wilson & Fazenda, 2016; Mourgela et al., 2024; Deruty & Tardieu, 2014). For instance, an enthusiast recording vocals in a garage may introduce excessive reverberation, making the voice sound distant and "echoey." Similarly, using inexpensive microphones or misconfigured interfaces can lead to distortion and clipping when loud peaks exceed the recording range, resulting in harsh crackles or flattened dynamics (Zang et al., 2025). Tonal imbalances are also common: a home recording might sound overly "muddy" or "tinny" if certain frequency bands dominate or vanish due to poor room acoustics or improper microphone placement. Even the stereo image can be narrowed or skewed, reducing the sense of space in the mix. In practice, engineers address these problems with specialized tools: e.g., dereverberation plugins to remove room echo, declipping algorithms to reconstruct saturated peaks, equalizers to rebalance frequencies, and stereo enhancers to widen the image. Mastering a flawed track has become a labor-intensive process requiring expert skill and multiple stages of manual adjustment. The need for an automated all-in-one solution is evident. Creators with limited resources often lack the expertise to apply the right combination of restoration tools, and a piecemeal approach may fail to fully recover a track's fidelity. This motivates *SonicMaster*, a unified approach to music restoration and mastering that can correct a broad spectrum of audio degradations within a single model.

With the rise of LLMs and language-driven interfaces, natural-language control is becoming increasingly important in audio production systems. However, music-production terminology is inherently ambiguous and highly subjective: the same perceptual attribute may be described differently by listeners, musicians, and engineers, while identical terms can carry different meanings across production contexts. Prior work has examined this semantic ambiguity in music production language as well as the influence of technological advancement on practices and language (Moffat et al., 2022; Wilmering et al., 2013; Pras et al., 2013); addressing it comprehensively, however, is beyond the scope of this paper.

---

[1]Singapore University of Technology and Design, Singapore
[2]Ca' Foscari University of Venice, Italy. Correspondence to: Jan Melechovsky <jan_melechovsky@mymail.sutd.edu.sg>.

*Proceedings of the 43$^{rd}$ International Conference on Machine Learning*, Seoul, South Korea. PMLR 306, 2026. Copyright 2026 by the author(s).

The distinction between "mixing" and "mastering" is similarly fluid. Mixing is commonly understood as the process of editing, balancing, and combining individual instrument tracks into a coherent composition, whereas mastering broadly refers to perceptual enhancement and final optimization of music for distribution (e.g., album-level cohesion or format-specific preparation such as vinyl mastering). In practice, these stages often overlap, particularly in modern digital production workflows. In this work, we use the term mastering to denote targeted perceptual enhancement of degraded music recordings toward professionally polished quality (including certain mixing tasks), rather than the full scope of traditional professional mastering. Given that this work is first of its kind, the current focus is limited to single-song processing rather than album-level mastering for cross-track cohesion.

We introduce a flow-based generative framework (Liu et al., 2023b; Esser et al., 2024) that simultaneously performs dereverberation, equalization, declipping, dynamic-range expansion, and stereo enhancement. The backbone is trained on a curated corpus of polyphonic music rendered through a combinatorial grid of simulated degradations, enabling the network to learn the joint statistics and cross-couplings of common artifacts rather than treating them in isolation. This joint training obviates error-prone cascades of task-specific modules and reduces inference to a single forward pass.

Crucially, *SonicMaster* incorporates multimodal conditioning through natural language instructions that capture production objectives. A prompt such as `reduce the hollow room sound` attenuates late reflections without suppressing desirable early reverberation, whereas `increase the brightness` selectively enhances the treble frequencies while preserving spectral balance elsewhere.In the absence of a prompt, *SonicMaster* switches to an automatic mode that applies perceptually balanced mastering. Existing speech restoration models (e.g. VoiceFixer by Liu et al. (2021)) also address artifacts sequentially, ignoring their mutual influence. By unifying restoration and mastering tasks under a single, prompt-driven generative model, *SonicMaster* delivers professional-grade improvements while affording fine-grained creative control. Recent advances such as Mustango (Text-guided music generation) by Melechovsky et al. (2024), FlowSep by Yuan et al. (2025) (text-guided source separation), TangoFlux by Hung et al. (2024) (reward-optimized text-to-audio diffusion), and instruction-guided models like AUDIT (Wang et al., 2023) or AudioLDM/AudioLDM2 (Liu et al., 2023a; 2024) illustrate powerful generative methods, but they target orthogonal tasks—generation, separation, or localized editing—rather than unified restoration. In contrast, *SonicMaster* uniquely addresses comprehensive multi-artifact music restoration and mastering through a single controllable rectified-flow architecture, unifying dereverberation, declip-

ping, tonal rebalancing, dynamics, and stereo enhancement under prompt guidance.

In the absence of text-conditioned music-restoration data, we build a new large-scale corpus for controllable restoration. From $\approx 580\,k$ Jamendo recordings, we retain $\approx 25\,k$ high-quality 30-s segments, balanced across 10 genre groups by production quality score. Each clean clip is corrupted with one to three of 19 common effects drawn from five categories: EQ, dynamics, reverb, amplitude, and stereo—producing paired degraded versions. Every degraded sample is accompanied by a natural-language prompt describing the artifact or required fix, and all random effect parameters are stored as metadata. Ultimately, *SonicMaster* aims to democratize professional-grade sonics, ensuring that the quality of a musical idea is no longer gatekept by the quality of the user's recording environment or technical proficiency. Our main contributions are as follows:

- We introduce *SonicMaster*, the first *flow-matching* model to simultaneously address 19 common degradations, including reverb, EQ imbalance, clipping, dynamic range errors, and stereo artifacts in a *single* generative framework, eliminating sequential processing and cascading error.

- *SonicMaster* enables precise user control through natural language conditioning, allowing targeted corrections (e.g., `reduce hollow room sound` for dereverberation) while maintaining autonomous operation when prompts are unavailable, bridging automated and user-directed restoration paradigms.

- *SonicMaster* empowers musicians to translate subjective artistic intent into objective signal processing. By replacing complex parameters such as thresholds, ratios, etc. with semantic control, it enables creators to treat restoration as a creative extension of the recording process rather than a corrective chore.

- We construct and release[1] the first text-conditioned music-restoration corpus: 25k high-fidelity Jamendo segments spanning 10 genres, each paired with 7 degraded versions, detailed metadata, and a natural-language instruction describing the required fix, resulting in 175k audio pairs.

## 2. Related Work

Restoring and mastering audio spans speech and music enhancement, audio inpainting, and source separation—areas that have mostly been handled separately (Záviška et al., 2020). Diffusion-based generative models and text-guided

---

[1]`https://github.com/AMAAI-Lab/SonicMaster`

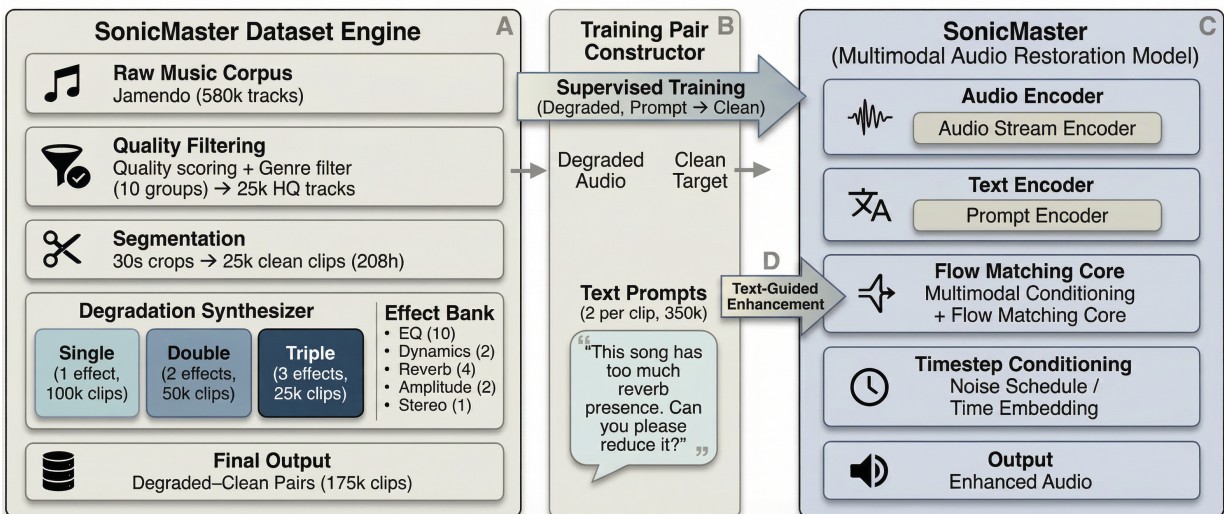

*Figure 1. SonicMaster* system overview. From left to right: dataset creation via controlled degradations, supervised training pair construction with text prompts, and the multimodal flow matching based *SonicMaster* model for text-guided audio enhancement.

audio editing (Hou et al., 2025; Jiang et al., 2025; Zhang et al., 2024; Manor & Michaeli, 2024) now let us tackle these problems together. We review these advances and the gaps that *SonicMaster* aims to fill. Early audio restoration efforts typically focused on single domains or isolated tasks, addressing issues like noise, clipping, or reverb in separation. Speech enhancement (Yousif & Mahmmod, 2025) and music enhancement evolved largely independently, and tasks such as audio inpainting or source separation were treated with specialized methods (Lemercier et al., 2025).

**Audio Inpainting, Mixing and Declipping:** Early signal-based methods addressed only minor gaps ($< 10$ ms), leaving longer dropouts unresolved. Recent deep generative approaches, particularly diffusion, enable realistic long-form inpainting and declipping (Moliner & Välimäki, 2024). Instruction-guided diffusion further introduces controllability (Wang et al., 2023; Liu et al., 2023a), while VoiceFixer (Liu et al., 2021) performs joint speech restoration without user control. In music, neural frameworks now address distortion removal (Imort et al., 2022), effect-chain recovery (Lee et al., 2024), and iterative or differentiable mixing (Steinmetz et al., 2021; Bhandari et al., 2025; Li et al., 2024), reflecting a shift toward unified, interpretable restoration.

**Equalization and Tonal Restoration:** Learning-based equalization remains relatively underexplored. Mockenhaupt et al. (2024) use CNNs to predict parametric settings for stems, surpassing traditional heuristics. Other frameworks address spectral balance implicitly through restoration, such as bandwidth extension in VoiceFixer (Liu et al., 2021) or treating imbalance as a degradation artifact in MaskSR (Li et al., 2024). For interactive control, Text2FX (Chu et al., 2025) leverages CLAP embeddings (Elizalde

et al., 2023) to map natural language prompts directly to EQ and reverb parameters.

## 3. Method

### 3.1. Dataset

In the absence of text-conditioned music restoration datasets, we generate *SonicMaster* dataset by pairing high-quality audio with systematically applied degradations and corresponding natural language instructions. We source 580K recordings from Roy et al. (2025) and Jamendo API [2]. We ensure balanced genre representation by defining 10 groups, where each group consists of multiple semantically related genre tags, e.g., Hip-Hop genre group containing the following tags: "rap", "hiphop", "trap", "alternativehiphop", "gangstarap". Complete taxonomies are provided in the Appendix. Using the Audiobox Aesthetics toolbox (Tjandra et al., 2025), we selected 2 500 songs per group via adaptive quality thresholds (6.5–8) and extracted random 30s excerpts (from 15%–85% of track duration). The complete pipeline is illustrated in Figure 1. We train *SonicMaster* by applying 19 distinct degradations to the audio and pairing each with a matching natural language editing instruction. The degradations span five classes: (i) EQ, (ii) Dynamics, (iii) Reverb, (iv) Amplitude, and (v) Stereo.

**Equalization (EQ):** Spectral degradations cover 10 effects targeting perceptual characteristics: Brightness, Darkness, Airiness, Boominess, Muddiness, Warmth, Vocals, Clarity, Microphone, and X-band. Brightness, Darkness, Airiness, Boominess, and Warmth are emulated with low- or high-shelf EQ; Clarity with a Butterworth low-pass filter; Vocals

---

[2]https://www.jamendo.com/

and Muddiness with Chebyshev-II band-pass filters. Microphone applies one of 20 Poliphone transfer functions (Salvi et al., 2025), while X-band uses an 8–12-band, logarithmically spaced peaking EQ with 6 dB gain per band.

**Dynamics:** Temporal envelope modification via two functions: Compression (feedforward dynamic range compression) and Punch (transient shaping). Both exhibit lossy, non-invertible characteristics, rendering exact restoration mathematically ill-posed, requiring learned approximations.

**Reverb:** The Reverb category contains four distinct approaches: three of them utilise the Pyroomacoustics library (Scheibler et al., 2018), which simulates acoustic environments with the image source method. We simulate three types of rooms: Small, Big, and Mixed. For our fourth Reverb function, we utilize 12 selected room impulse responses from the openAIR library dataset (Howard & Angus, n.d.), which give us audio with more real-life properties. The resulting impulse responses from all the functions are convoluted with the clean signals.

**Amplitude:** Two complementary degradations target signal amplitude: Clipping/Volume. Clipping introduces hard nonlinear distortion by constraining peak amplitudes to predefined thresholds; Volume reduction attenuates signals to near-inaudible levels, degrading the signal-to-quantization-noise ratio and simulating poor recording practices.

**Stereo:** A function to de-stereo the audio recording – tracks undergo stereo content analysis via left-right channel difference standard deviation (threshold: 0.08); qualifying recordings are converted to monophonic by channel summation, simulating poor mixing or playback equipment limitations.

Each ground truth yields 7 corrupted variants: 4 with a single, 2 with double, and 1 with triple degradation. In multi-degradation, we sample at most one effect from each of the 5 categories, so an EQ choice, for instance, blocks further EQ picks. To avoid duplicates in the single-degradation set, high-probability effects with narrow parameter ranges: Stereo, Clipping, and Punch, are used only once (in the 4 versions per original, e.g., there cannot be two single-degraded versions with Stereo degradation, as they would be identical). Each degradation is linked to a one-sentence instruction from 8–10 possible options (all written by a music expert); these sentences are concatenated into the full prompt, and we store two prompt variants per clip for robustness. We also record every applied effect and its parameters (gain, absorption), supporting tasks such as parameter prediction. For Compression and Reverb, there is a 15% chance of injecting "hidden clipping" with no corresponding instruction to emulate real life cases of constructive interference in a reverberant room, or overcompensated gain setting of a compressor. When neither hidden clipping nor an Amplitude effect is present, the audio is peak-normalised

to a random level between $0.8 - 1.0$. Further details can be found in Appendix.

### 3.2. *SonicMaster* Architecture

*SonicMaster* employs a hybrid architecture combining Multimodal Diffusion Transformer (MM-DiT) (Esser et al., 2024) blocks with subsequent Diffusion Transformers (DiT) layers (Peebles & Xie, 2023). The stereo waveforms sampled at 44.1 kHz undergo VAE encoding (Evans et al., 2024) into compact spectro-temporal latent representations. Restoration therefore occurs entirely in this learned space, allowing large receptive fields without sample-level overhead. Let $y_{\text{deg}}, y_{\text{clean}} \in \mathbb{R}^{B \times 2 \times T}$ denote degraded and clean stereo waveforms, respectively. A frozen VAE encoder $E_\phi$ maps waveforms into latent tensors: $x = E_\phi(y) \in \mathbb{R}^{B \times C \times F \times T}$, where $C$ is the latent channel dimension, $F$ the frequency axis, and $T$ the temporal axis. Paired training samples are defined as

$$x_1 = E_\phi(y_{\text{deg}}), \qquad x_0 = E_\phi(y_{\text{clean}}). \qquad (1)$$

The MM-DiT processes degraded latent representations alongside text embeddings from a frozen FLAN-T5 encoder (Chung et al., 2024). The resulting conditioned representations pass through subsequent DiT layers to predict the flow velocity $v_t$, steering the latent toward its clean target. Prompts such as "reduce reverb" bias this prediction trajectory to suppress decay tails, while the downstream DiT layers refine musical coherence.

**Audio and Text Encoding:** We adopt the Stable Audio Open VAE (Evans et al., 2024) to encode–decode stereo signals sampled at 44.1 kHz, yielding a compact latent representation while retaining high-fidelity reconstruction. Text instructions are embedded with FLAN-T5 Large (Chung et al., 2024); the resulting tensor

$$c_{\text{text}} \in \mathbb{R}^{B \times S_{\text{text}} \times D_{\text{text}}}, \quad D_{\text{text}} = 1024, \qquad (2)$$

is used as a conditioning signal.

During training, we employ classifier-free guidance by randomly dropping the text conditioning with probability $p_{\text{cfg}}$. At inference, we use *full guidance only* and directly predict

$$\hat{v}_t = f_\theta(x_t, t, c_{\text{text}}), \qquad (3)$$

without guidance interpolation. This enforces strict adherence to the user prompt; when no prompt is provided, conditioning on $c_\varnothing$ naturally yields an unguided prediction.

**Pooled-Audio Conditioning:** A pooled-audio branch, active in 25% of training cases, concatenates a temporally averaged 5–15s clean cue with the pooled prompt embedding and injects it at every MM-DiT/DiT layer, enabling seamless

chaining of 30s segments for long-form generation while degrading gracefully when no reference is supplied. Formally, let $y_{\text{cue}}$ denote the optional clean audio reference segment. Its latent representation is $x_{\text{cue}} = E_\phi(y_{\text{cue}})$, which is temporally pooled to obtain: $\bar{x}_{\text{cue}} = \text{Pool}_t(x_{\text{cue}}) \in \mathbb{R}^{B \times C \times F}$. Similarly, the text embedding is pooled along the sequence axis as $\bar{c}_{\text{text}} = \text{Pool}_s(c_{\text{text}}) \in \mathbb{R}^{B \times D_{\text{text}}}$. These pooled representations are concatenated and projected to form a global conditioning vector

$$u = W_u\big[\text{vec}(\bar{x}_{\text{cue}}); \bar{c}_{\text{text}}\big] \in \mathbb{R}^{B \times d_u}, \qquad (4)$$

which is injected into every MM-DiT and DiT block.

**Rectified Flow Training:** *SonicMaster* employs rectified flow (Liu et al., 2023b; Esser et al., 2024) to predict flow velocity from degraded to clean audio in latent space, unlike diffusion models that map noise to output distributions (Fei et al., 2024; Hung et al., 2024). We assign timestep $t = 1$ to the degraded latent $x_1$ and $t = 0$ to the clean latent $x_0$. During training, we sample

$$x_t = tx_1 + (1 - t)x_0, \qquad (5)$$

where $t$ is drawn from a skewed distribution $p(t) = 0.5U(t) + t$, $t \in [0, 1]$, emphasizing highly degraded inputs. The model is trained to predict the flow velocity $v_t = -\frac{dx_t}{dt} = x_0 - x_1$. A neural network $f_\theta$ estimates the velocity $\hat{v}_t = f_\theta(x_t, t, c_{\text{text}})$, where $c_{\text{text}}$ is injected both as a token stream in MM-DiT blocks and as a pooled conditioning signal via adaptive layer normalization (AdaLN). The training objective for *SonicMaster* is

$$L(\theta) = \mathbb{E}_{t,x_1,x_0}\big\| f_\theta(x_t, t, c_{\text{text}}) - v_t \big\|_2^2. \qquad (6)$$

**MM-DiT Dual-Stream Fusion:** Let $Z_x \in \mathbb{R}^{B \times N_x \times D}$ denote latent tokens derived from $x_t$, and $Z_c \in \mathbb{R}^{B \times S_{\text{text}} \times D}$ denote text tokens. A single MM-DiT block performs cross-attention and feed-forward updates as

$$Z_x' = Z_x + \text{Attn}(Q = Z_x, K = Z_c, V = Z_c), \quad (7)$$
$$Z_x'' = Z_x' + \text{MLP}\big(\text{AdaLN}(Z_x'; u)\big), \qquad (8)$$

with $Z_x''$ passed to subsequent MM-DiT or DiT layers.

**Inference:** During inference, *SonicMaster* takes in a degraded audio input and a text instruction to perform the desired restoration or mastering operation. Inference is also possible without text input in the so-called auto-correction mode. Inference follows the probability flow ODE

$$\frac{dx(t)}{dt} = f_\theta(x(t), t, c_{\text{text}}), \quad x(1) = x_1, \qquad (9)$$

whose solution at $t = 0$ yields the restored latent $\hat{x}_0$. We adopt a forward Euler solver for efficiency: $x_{t-h} = x_t + h\hat{v}_t$, where $h \in [0, 1]$ is the integration step size.

**Long-Form Generation:** To process full-length songs, *SonicMaster* operates on chunks of 30s and then connects the segments together. After the first segment is inferred, the last 10s of its output are used to condition the next segment through the pooled-audio branch. Let $\hat{y}^{(n)}$ and $\hat{y}^{(n+1)}$ denote two consecutive decoded segments with overlap length $L$. The merged waveform $\tilde{y}$ is computed via overlap–add:

$$\tilde{y}[t] = \begin{cases} \hat{y}^{(n)}[t], & t < T - L, \\ (1 - w[t - (T - L)])\hat{y}^{(n)}[t] + \\ w[t - (T - L)]\hat{y}^{(n+1)}[t - (T - L)], & t \in [T - L, T), \end{cases} \qquad (10)$$

where $w \in [0, 1]^L$ is a linear cross-fade window.

# 4. Experimental Setup and Baselines

## 4.1. Baselines and Training Setup

We train *SonicMaster* using 5 NVIDIA L40S GPUs for 40 epochs with a total batch size of 80. We adopt classifier-free guidance (Ho & Salimans, 2022) by (i) dropping the text prompt in 10% of samples and (ii) replacing it in another 10% with one of four generic phrases ("Make it sound better!", "Master this track for me, please!", "Improve this!", "Can you improve the sound of this song?"). In 25% of cases, the model is additionally conditioned via the pooling branch on the first 10 sec of clean audio. Unless stated otherwise, all experiments follow these conditioning settings while comparing *SonicMaster* variants and baselines.

We compare against recent approaches, alongside ablation studies for different *SonicMaster* configurations:: (i) **Degraded input**—the original corrupted audio; (ii) **Reconstructed input**—the same audio passed through the VAE encoder–decoder; (iii) **Text2FX-EQ**, an EQ baseline using Text2FX$_{cos}$ and Text2FX$_{dir}$ (Chu et al., 2025) with 600 iterations and a 0.01 learning rate to correct EQ degradations via our prompts; (iv) **WPE** dereverberation, the Weighted Prediction Error algorithm (Nakatani et al., 2010) with a prediction order of 30; (v) **HPSS** dereverberation, harmonic–percussive source separation (`librosa.decompose.hpss`) with 6 dB and 12 dB harmonic attenuation; (vi) `Mel2Mel` + `DiffWave` (Kandpal et al., 2022) framework that treats mel-spectrogram enhancement as an image-to-image translation followed by diffusion vocoding for music restoration. and (vii) three *SonicMaster* variants—*SonicMaster*$_{Small}$ (2 MM-DiT + 6 DiT), *SonicMaster*$_{Medium}$ (4 MM-DiT + 12 DiT *or* 6 MM-DiT + 6 DiT), and *SonicMaster*$_{Large}$ (6 MM-DiT + 18 DiT). Given that Text2FX[3] is not a restoration model, we further deploy its directional variant as a meaningful text-guided audio mastering baseline. Text2FX-directional (Text2FX$_{dir}$) is specifically designed for instruction-following tasks: it

---

[3]Appendix A.3 has details of the Text2FX-directional, both loss formulations and EQ prompt construction.

steers the audio embedding in the same semantic direction defined by a target prompt and its contrast prompt.

Evaluation follows two axes: (i) global perceptual fidelity, measured using FAD on CLAP embeddings (Elizalde et al., 2023), KL divergence, SSIM on 128-bin mel-spectrograms, and the Production Quality (PQ) score from Audiobox Aesthetics (Tjandra et al., 2025); and (ii) degradation-specific restoration efficacy, quantified as average absolute error reduction by comparing each degraded clip in a 7k-sample test set to its clean reference (1k samples) before and after *SonicMaster* processing, with relative decreases indicating proximity to ground truth.

For X-band EQ and microphone-TF degradations, we compute the spectral balance over nine frequency bands and report their cosine distance. All other EQ effects are scored by the energy ratio between the affected band and the full spectrum. Compression is measured as the standard deviation of frame-level RMS (2048-sample frames, 1024 hop); punch as the mean onset-envelope value (`librosa.onset.onset_strength`). Because RT60 estimates are unreliable on dense mixes, reverb is assessed via the Euclidean distance of modulation spectra. Clipping uses spectral flatness; volume, the global RMS; and stereo width, the RMS ratio of the mid and side signals, $\mathrm{RMS}\left[\frac{L-R}{2}\right]/\mathrm{RMS}\left[\frac{L+R}{2}\right]$. We report the average absolute error value (GT vs inferred sample) of all the metrics except where mentioned differently (X-band, microphone-TF, and reverb). Each metric is described in Appendix A.5.

Listeners evaluated 43 audio sample pairs comprising degraded inputs and corresponding *SonicMaster* outputs: 38 single-degradation samples (2 per degradation type), 3 double-degradation, and 2 triple-degradation cases. Using a 7-point Likert scale, participants rated text relevance, input audio quality (Quality1), output audio quality (Quality2), output consistency, and general preference (whether input or output sample is preferred in terms of listener enjoyment and satisfaction on the 7-point scale, with 1 meaning full preference of the input sample and 7 of the output sample). The study involved 12 listeners, including 7 music experts and 5 Music Information Retrieval researchers. We provide an example of the listening study in Appendix A.7. Furthermore, to benchmark against existing methods, we conducted an additional study with 20 participants comparing *SonicMaster* against Text2FX$_{cos}$ and Text2FX$_{dir}$ (Chu et al., 2025), and Mel2Mel + Diffwave (Kandpal et al., 2022) on 20 randomly selected samples from our test set. The evaluation included 10 samples with X-band EQ degradation and 10 with reverberation artifacts. Note that Text2FX$_{cos}$ and Text2FX$_{dir}$ are limited to EQ effects as their reverb effect is only additive, thus excluded. Since the baseline method's evaluation sets are not publicly available, we performed this comparison exclusively on our curated test data.

## 5. Results

### 5.1. Objective Evaluation

**Degradation-Specific Performance:** Tables 1 and 2 demonstrates *SonicMaster*'s superiority over baselines of Text2FX in EQ, and WPE/HPSS in Reverb. *SonicMaster* improves in all categories when compared to the degraded and reconstructed inputs. Furthermore, the reconstructed input metrics are overall slightly worse (with exceptions) than those of the ground truth degraded inputs.

**Perceptual Quality Assessment:** Table 3 reveals *SonicMaster* outperforms the degraded inputs in both PQ and KL. FAD is marginally higher than that of the degraded audio, yet markedly lower than the reconstructed baseline. Furthermore, *SonicMaster* achieves a significant increase in PQ, almost reaching the level of ground truth mastered reference. In SSIM, *SonicMaster* exhibits lower scores than degraded inputs but achieves superior performance compared to the reconstruction baseline.

Full-song evaluations further validate *SonicMaster*'s effectiveness, showing consistent improvements in EQ-related metrics (Table 1) and across most degradation types (Table 2). Objective fidelity metrics (Table 3) show reduced SSIM and FAD relative to degraded inputs, except for FAD under multi-degradation settings, highlighting *SonicMaster*'s ability to handle compound degradations.

**Comparison with Removal Models:** Figure 2 illustrates the distinction between effect removal and text-guided mastering. While models such as DPTNet (Chen et al., 2020), UMX (Stöter et al., 2019), DCUNet (Choi et al., 2018), TCN (Rethage et al., 2018; Steinmetz & Reiss, 2022), and HDemucs (Défossez, 2021) focus on effect removal with minimal alteration (Rice et al., 2023) (best baseline: 20.08 dB for Dynamics, 13.59 dB for Reverb), *SonicMaster* performs text-guided mastering that applies intentional tonal and dynamic shaping. All baselines are trained following the RemFX protocol (Rice et al., 2023) using effect-specific supervision with L1 + multi-resolution STFT losses, and evaluated on the official test split containing clean vs. effected pairs for each degradation type. We test *SonicMaster* on the same test set, focusing on the two degradations: Dynamics and Reverb used in Rice et al. (2023). This broader objective enables *SonicMaster* to reconstruct a more coherent musical structure, achieving substantially higher SI-SDR scores of 47.11 dB (Dynamics) and 45.76 dB (Reverb).

### 5.2. Ablation Studies

Ablation studies indicate that *SonicMaster* is robust to architectural scaling, ODE solver choice, and variations in audio conditioning length and presence, with largely stable performance across settings (see Appendix A.9). Removing text conditioning preserves overall audio quality but reduces

| Model | Clarity | Boom | Airy | Bright | Dark | Muddy | Warm | Vocals | Mic. | X-band |
|---|---|---|---|---|---|---|---|---|---|---|
| *Snippet Evaluation (Short Segments)* | | | | | | | | | | |
| Degraded Input | 0.0238 | 0.3601 | 0.0049 | 0.0143 | 0.0893 | 0.4560 | 0.4345 | 0.2525 | 0.2393 | 0.1782 |
| Reconstructed Input | 0.0243 | 0.3717 | 0.0051 | 0.0151 | 0.0728 | 0.4749 | 0.4456 | 0.2525 | 0.2379 | 0.1854 |
| Mel2Mel + Diffwave (Kandpal et al., 2022) | 0.0278 | 0.3561 | 0.0049 | 0.0135 | 0.0855 | 0.4705 | 0.4436 | 0.2560 | 0.2604 | 0.1885 |
| Text2FX$_{cos}$ (Chu et al., 2025) | 0.0219 | 0.3809 | 0.0055 | 0.0276 | 0.2112 | 0.3651 | 0.4955 | 0.2199 | 0.4441 | 0.3419 |
| Text2FX$_{dir}$ (Chu et al., 2025) | 0.0421 | 0.3977 | 0.0206 | 0.0143 | 0.3021 | 0.2602 | 0.5461 | 0.2517 | 0.6120 | 0.5038 |
| *SonicMaster* (Ours) | **0.0114** | **0.0834** | **0.0019** | **0.0059** | **0.0058** | **0.0388** | **0.0617** | **0.0576** | **0.0088** | **0.0358** |
| *Full Song Evaluation (Long-Form)* | | | | | | | | | | |
| Degraded Input | 0.0290 | 0.3231 | 0.0048 | 0.0124 | 0.0983 | 0.4606 | 0.4810 | 0.2274 | 0.2403 | 0.1737 |
| *SonicMaster* (Ours) | **0.0102** | **0.0639** | **0.0021** | **0.0060** | **0.0065** | **0.0329** | **0.0510** | **0.0517** | **0.0070** | **0.0289** |

*Table 1.* EQ Objective Evaluation (Average Absolute Error). **Bold** = best performance (lowest error).

| Model | Reverb | | | | Dynamics | | Amplitude | | Stereo |
|---|---|---|---|---|---|---|---|---|---|
| | Small | Big | Mix | Real | Comp. | Punch | Clip | Vol. | |
| *Snippet Evaluation (Short Segments)* | | | | | | | | | |
| Degraded Input | 0.4457 | 0.4243 | 0.5045 | 0.4639 | 0.0496 | 0.1200 | 5.122 | 0.1813 | 0.4183 |
| Reconstructed Input | 0.4686 | 0.4507 | 0.5433 | 0.4908 | 0.0494 | 0.0590 | 3.871 | 0.1810 | 0.4181 |
| HPSS 6 dB | 0.4419 | 0.4240 | 0.4970 | 0.4537 | - | - | - | - | - |
| HPSS 12 dB | 0.4971 | 0.4739 | 0.5333 | 0.4814 | - | - | - | - | - |
| WPE (Nakatani et al., 2010) | 0.4849 | 0.4732 | 0.5207 | 0.4854 | - | - | - | - | - |
| Mel2Mel + Diffwave (Kandpal et al., 2022) | 0.4404 | 0.4387 | 0.4361 | 0.4368 | - | - | - | - | - |
| *SonicMaster* (Ours) | **0.3663** | **0.3726** | **0.3935** | **0.3109** | **0.0193** | **0.0871** | **1.506** | **0.0468** | **0.1058** |
| *Full Song Evaluation (Long-Form)* | | | | | | | | | |
| Degraded Input | 0.3667 | 0.3654 | 0.4706 | 0.3852 | 0.0598 | 0.1103 | 6.363 | 0.1829 | 0.4133 |
| *SonicMaster* (Ours) | **0.3954** | **0.4511** | **0.4191** | **0.4066** | **0.0258** | **0.1101** | **3.734** | **0.0424** | **0.0850** |

*Table 2.* Objective Scores: Reverb, Dynamics, Amplitude, Stereo. Clip scores multiplied by 1000. **Bold** = best performance (lowest error).

performance on targeted attributes, suggesting that text primarily enables controllable restoration rather than generic enhancement. Shuffling prompts at inference time further degrades targeted metrics while still outperforming the degraded input, confirming that text conditioning enforces semantic alignment. Differences across ablations are most pronounced in Clip, Stereo, and perceptual quality metrics.

### 5.3. Piano Recordings Evaluation

For testing generalization, we evaluate historical solo piano pieces[4] using established baselines: LTAS-EQ, BEHM-GAN (Moliner & Välimäki, 2023) model for bandwidth extension, and BABE/BABE-2 (Moliner et al., 2024; Moliner & Välimäki, 2024). BABE-2 represents a state-of-the-art specialized method for old recordings, it uses a diffusion prior to restore lost high frequencies and remove coloration, and has shown impressive improvements in archival music (Moliner et al., 2024). Despite lacking domain-specific training, *SonicMaster* came surprisingly close to these specialized baselines (Table 4). In objective evaluations, Son-

[4] http://research.spa.aalto.fi/publications/papers/dafx-babe2/

icMaster restored samples achieved a PQ of 6.93, nearly matching the 7.05 obtained by BABE-2.

### 5.4. Subjective Evaluation

Figure 3 shows results of the first listening study. Text relevance ratings are highest in the Amplitude (6.21), Stereo (5.75), and Reverb (5.36) categories, indicating effective declipping, volume increase, expansion of the stereo image, and dereverberation. These 3 categories also show the highest consistency and preference ratings. Dynamics and Amplitude show the biggest improvement in quality. EQ shows the fourth-best text relevance, but the worst preference ratings. This likely reflects the nature of some EQ effects being more stylistic or difficult to notice (e.g., airiness, boominess). Overall, *SonicMaster* samples are rated higher in quality compared to inputs and preferred across the board. A paired $t$-test on Quality1 and Quality2 shows statistically significant differences ($p < 0.05$ for Stereo, $p < 0.01$ for the rest) in all categories except EQ. Detailed results from the second listening study are reported in Appendix A.8 and consistently confirm *SonicMaster*'s preference over prior methods across both reverb and EQ degradations.

| Model | Single Deg. | | | | Double+Triple Deg. | | | | All | | | |
|---|---|---|---|---|---|---|---|---|---|---|---|---|
| | FAD↓ | KL↓ | SSIM↑ | PQ↑ | FAD↓ | KL↓ | SSIM↑ | PQ↑ | FAD↓ | KL↓ | SSIM↑ | PQ↑ |
| *Snippet Evaluation (Short Segments)* | | | | | | | | | | | | |
| GT Mastered Ref. | - | - | - | 7.886 | - | - | - | 7.886 | - | - | - | 7.886 |
| Degraded Input | 0.061 | 3.859 | **0.838** | 7.321 | 0.184 | 6.827 | **0.696** | 6.632 | 0.106 | 5.131 | **0.777** | 7.026 |
| Reconstructed Input | 0.139 | 3.990 | 0.574 | 7.172 | 0.290 | 6.984 | 0.507 | 6.501 | 0.196 | 5.273 | 0.546 | 6.885 |
| Mel2Mel + Diffwave (Kandpal et al., 2022) | 0.522 | 14.938 | 0.447 | 6.158 | 0.474 | 15.185 | 0.416 | 5.953 | 0.491 | 15.044 | 0.433 | 6.070 |
| *SonicMaster* (Ours) | **0.069** | **0.696** | 0.624 | **7.743** | **0.082** | **1.145** | 0.589 | **7.654** | **0.073** | **0.888** | 0.609 | **7.705** |
| *Full Song Evaluation (Long-Form)* | | | | | | | | | | | | |
| GT Mastered Ref. | - | - | - | 7.885 | - | - | - | 7.885 | - | - | - | 7.885 |
| Degraded Input | **0.087** | 2.937 | **0.834** | 7.325 | 0.223 | 5.679 | **0.682** | 6.606 | 0.142 | 4.308 | **0.758** | 6.965 |
| Reconstructed Input | 0.165 | 3.049 | 0.584 | 7.204 | 0.335 | 5.644 | 0.510 | 6.509 | 0.234 | 4.339 | 0.547 | 6.859 |
| *SonicMaster* (Ours) | 0.095 | **0.754** | 0.380 | **7.627** | **0.121** | **1.251** | 0.368 | **7.477** | **0.101** | **1.002** | 0.374 | **7.552** |

*Table 3.* Objective Scores: FAD (↓), KL (↓), SSIM (↑), and PQ (↑). KL values are multiplied by 1000 for readability. **Bold** indicates best performance (excluding ground truth reference).

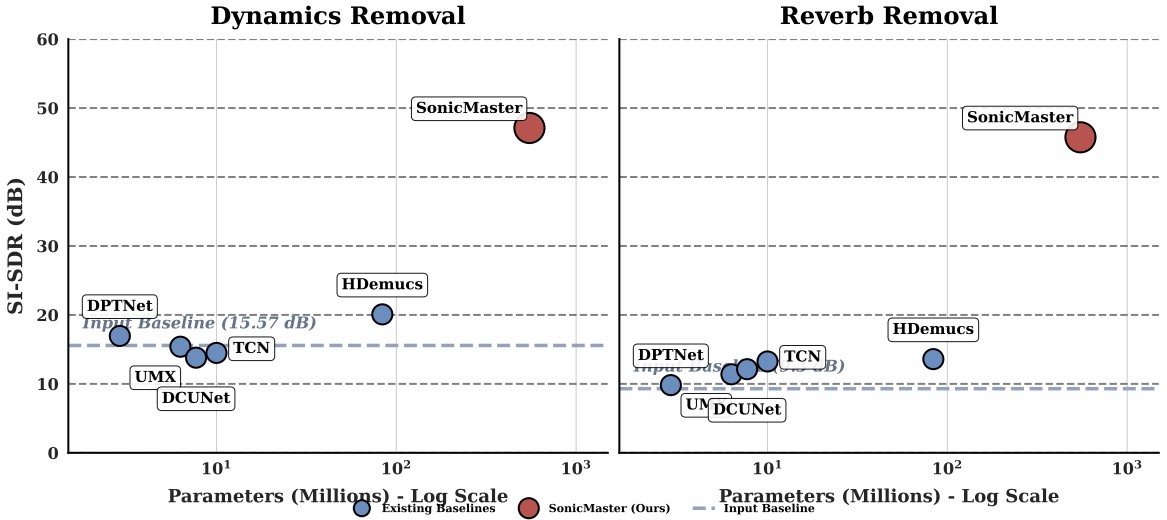

*Figure 2.* Comparison of SI-SDR scores (↑) for Dynamics and Reverb removal.

| Method | CE ↑ | CU ↑ | PC ↑ | PQ ↑ |
|---|---|---|---|---|
| Original | $6.94_{\pm 0.48}$ | $7.29_{\pm 0.43}$ | $3.45_{\pm 0.36}$ | $6.70_{\pm 0.50}$ |
| LTAS-EQ | $6.77_{\pm 0.54}$ | $7.04_{\pm 0.57}$ | $3.75_{\pm 0.45}$ | $6.49_{\pm 0.57}$ |
| BEHM-GAN | $6.82_{\pm 0.43}$ | $7.19_{\pm 0.44}$ | $3.47_{\pm 0.35}$ | $6.63_{\pm 0.56}$ |
| BABE | $\mathbf{6.96}_{\pm 0.37}$ | $\mathbf{7.32}_{\pm 0.37}$ | $3.32_{\pm 0.29}$ | $6.79_{\pm 0.36}$ |
| BABE-2 | $6.79_{\pm 0.34}$ | $7.16_{\pm 0.29}$ | $3.46_{\pm 0.28}$ | $\mathbf{7.05}_{\pm 0.27}$ |
| *SonicMaster* (ours) | $6.87_{\pm 0.55}$ | $7.25_{\pm 0.50}$ | $\mathbf{3.86}_{\pm 0.39}$ | $6.93_{\pm 0.52}$ |

*Table 4.* Comparison of mean scores across CE, CU, PC, and PQ.

## 6. Discussion

Experiments confirm that *SonicMaster*'s generative approach is effective when trained on a large corpus with a suitable objective. The historical piano experiment demonstrated *SonicMaster*'s strong generalization: even on out-of-domain, severely degraded audio, it produced enhancements

close to the best specialized solution, BABE-2. This highlights the potential of general-purpose audio restoration AI. However, a key limitation is that the lossy latent representation can introduce artifacts, such as robotic vocals or muted instruments, especially in certain genres. Evaluating reverberation in dense music is challenging, and how *SonicMaster* removes it in latent space is not explicitly observable, making metric selection difficult. A deeper study of this issue would benefit the community. In addition, while this work dealt with 19 different degradations, the list is far from exhaustive. The language used to describe the desired change can be further expanded by more musical jargon. Feasibility studies of the chosen language as well as the text-control interface present a valuable avenue for future work. Last but not least, this work deals with mastering in a narrow sense, focusing on single-song improvement (for

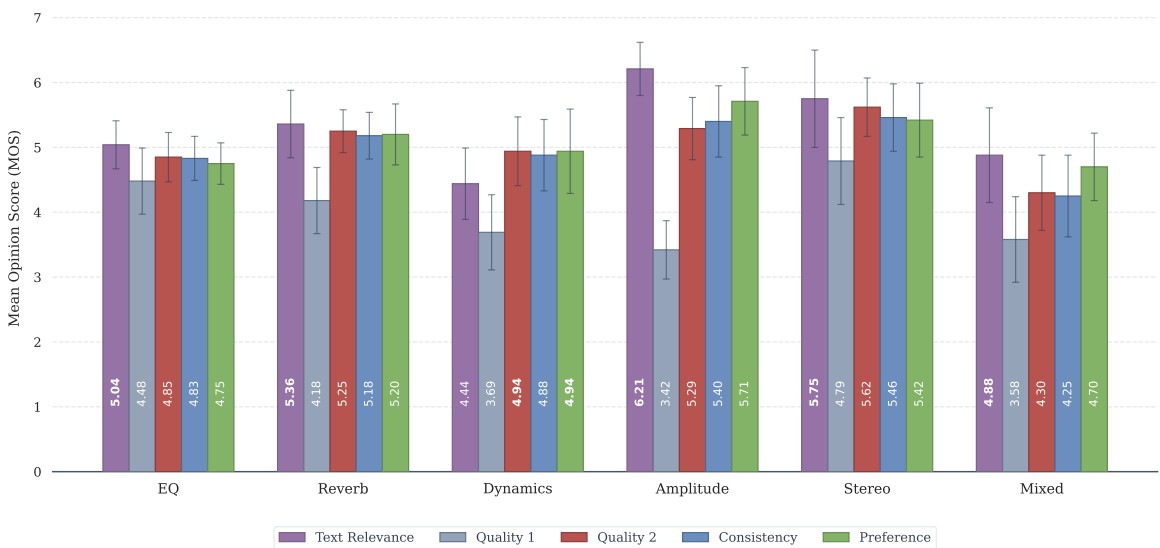

*Figure 3.* Listening study - *SonicMaster*'s performance on specific degradations – MOS 95% CI).

amateur musicians), but omitting broader concepts such as album-level editing for cross-track cohesion. These areas present avenues for future work. We provide a full list of limitations in Appendix A.10.

## 7. Conclusion

We introduced *SonicMaster*, the first unified text-guided generative model for music restoration and mastering, capable of handling 19 diverse degradations within a single framework. Our contributions further include the creation of a paired degraded–clean dataset with textual annotations, the introduction of a flow-matching paradigm for directly learning restoration mappings, and the integration of natural language conditioning for precise and flexible control. Evaluations show that *SonicMaster* consistently improves the audio quality, outperforming baselines in terms of objective metrics and listener studies. It also achieved strong zero-shot performance on old piano recordings, highlighting its versatility and suggesting a path toward a generalist restoration framework–one capable of addressing diverse challenges through prompt guidance while approaching the quality of specialist methods.

## Acknowledgements

This work has received support from SUTD's Kickstart Initiative under grant number SKI 2021 04 06 and MOE Tier 2 under grant number MOE-T2EP20124-0014.

We thank Christopher Johann Clarke and Jer-Ming Chen for their valuable insights during the development of this work.

## Impact Statement

This paper presents work whose objective is to advance the field of AI for music restoration and mastering. The proposed methods are intended to improve the efficiency and accessibility of music enhancement workflows.

The societal and ethical implications of this work are consistent with those of prior research in AI based music processing. Potential benefits include lowering the technical barrier to improving the quality of music recordings in non-professional or resource-constrained settings. Potential risks include increased competition and reduced demand in the music production and mastering field. However, we believe that human expertise is still far from being replaced and is currently invaluable, even more so given the subjective aspect of music and both specialized expertise of professionals and endlessly diverse needs of musicians and producers.

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

# A. Appendix

## A.1. The Use of Large Language Models

We employed a Large Language Model to assist with reducing wordy paragraphs to help the paper fit in the page limit.

## A.2. Genre tags

We grouped genre tags into genre groups, as depicted in Table 5. Each row links a coarse "Group" label—such as Rock, Electronic, or Jazz/Blues—to the fine-grained "Genre tags" that appear in the metadata. These tags enumerate substyles (e.g., `progressiverock`, `deephouse`, `acidjazz`), which allows us to aggregate diverse representations inside each of the genre groups.

*Table 5.* Genre groupings by metadata tags used in our dataset.

| Group | Genre tags |
|---|---|
| Rock | rock, alternativerock, poprock, classicrock, hardrock, progressiverock, stoner, psychedelicrock, garage, indierock |
| Pop | pop, electropop, dancepop, dance, alternativepop, adultcontemporary, indiepop |
| Electronic | electronic, house, techno, trance, edm, electrohouse, deephouse, progressivehouse, electroswing, synthwave, electronica |
| Hip-Hop | rap, hiphop, trap, alternativehiphop, gangstarap |
| Folk | folk, singersongwriter, americana, country, bluegrass, folklore |
| Metal | metal, deathmetal, blackmetal, thrashmetal, heavymetal, numetal, metalcore, hardcore, alternativemetal, doommetal |
| World | world, latin, reggaeton, afrobeat, african, indian, oriental, celtic, salsa, flamenco, jpop, middleeastern, asian, reggae |
| Jazz/Blues | jazz, blues, funk, acidjazz, jazzfusion, smoothjazz, jazzfunk, soul, swing, rnb, alternativernb |
| Chill | ambient, downtempo, chillout, chillhop, lofi, newage, darkambient, triphop, chillwave, idm, dreampop |
| Classical | classical, filmscore, neoclassical, symphonic, opera, baroque, medieval, avantgarde, production, choral |

## A.3. Text2FX-Directional Baseline for the EQ Task

For the equalization (EQ) experiments, we include the Text2FX-Directional method (Chu et al., 2025) as a text-guided audio transformation baseline. Although Text2FX is not a restoration model, SonicMaster is instruction-conditioned; therefore, a text-conditioned FX optimizer offers a meaningful point of comparison for evaluating how well different systems follow natural-language EQ instructions.

### A.3.1. DIRECTIONAL LOSS FORMULATION

Text2FX-Directional uses CLAP audio/text embeddings to align the *change in audio embedding* with the *semantic direction* defined by a target prompt and a contrast prompt. Let $f_a$ and $f_t$ denote the CLAP audio and text encoders, and let $g(x; \theta)$ be a differentiable 6-band parametric EQ (dasp-pytorch). Given degraded audio $x_{\text{deg}}$ and prompts $t_1$ (contrast) and $t_2$ (target), we define:

$$A_1 = f_a(x_{\text{deg}}),$$
$$A_2(\theta) = f_a\big(g(x_{\text{deg}}; \theta)\big),$$
$$T_1 = f_t(t_1),$$
$$T_2 = f_t(t_2).$$

The method encourages the audio embedding to move from $A_1$ to $A_2$ in the same direction as the text embedding moves from $T_1$ to $T_2$. Let

$$d_a(\theta) = \frac{A_2(\theta) - A_1}{\|A_2(\theta) - A_1\|_2}, \qquad d_t = \frac{T_2 - T_1}{\|T_2 - T_1\|_2}.$$

The directional loss is then:

$$\mathcal{L}_{\text{dir}}(\theta) = 1 - \cos\big(d_a(\theta), d_t\big).$$

We follow the optimization settings of (Chu et al., 2025): 600 Adam iterations (learning rate $1 \times 10^{-2}$), standard-normal parameter initialization, and a random circular time shift at each step to avoid fixation on audio content.

A.3.2. PROMPT AND CONTRAST-PROMPT CONSTRUCTION

Our EQ dataset contains natural-language instructions rather than the short adjectives used in (Chu et al., 2025). To maintain the $T_1 \rightarrow T_2$ structure required by the directional loss, we construct a semantically opposite *contrast prompt* for each instruction using GPT-4o with a constrained template ("write the opposite EQ action") and manual verification.

Examples used in our EQ evaluation include:

- **Clarity / Treble Boost:**
    - Target prompt ($T_2$): "Increase the clarity of this song by emphasizing treble frequencies."
    - Contrast prompt ($T_1$): "Decrease the clarity of this song by softening or reducing the treble frequencies and making it sound more dull and muffled."

- **Boominess / Low-End Enhancement:**
    - Target prompt ($T_2$): "Add weight and depth to the bottom end."
    - Contrast prompt ($T_1$): "Do the opposite of the following instruction: Add weight and depth to the bottom end."

- **Mic / Narrow-Band Coloration:**
    - Target prompt ($T_2$): "Balance the EQ, please."
    - Contrast prompt ($T_1$): "Do the opposite of the following instruction: Balance the EQ, please."

These pairs ensure that Text2FX-Directional receives properly opposed EQ semantics while matching the full-sentence instruction style of our enhancement dataset.

A.3.3. PURPOSE OF THIS BASELINE

Text2FX-Directional does not use the clean reference audio during optimization; thus it is *not* evaluated as a restoration model. Instead, we include it as a text-conditioned equalization baseline that evaluates: *How well can a CLAP-guided, single-instance EQ optimizer follow the same natural-language instructions given to SonicMaster?* This provides a fair, instruction-aligned comparison for EQ-specific transformations under identical textual guidance.

### A.4. Degradation functions

To create the SonicMaster dataset, we used a set of 19 degradation functions. The details of their implementation and parameter range are described in Table 6. Each of the groups, and subsequently each of the functions inside the groups, have their own probabilities/weights to be picked in our data creation pipeline. These are documented in Table 7.

**Peak normalisation of tracks:** In case of no intentional clipping, "hidden clipping", or a low volume degradation being used, all degraded versions of the SonicMaster dataset are normalised to a peak amplitude $y_{peak}$ drawn from a uniform distribution $y_{peak} \sim U(0.8, 1.0)$, track is then normalised as:

$$x_{norm} = \frac{x}{max(abs(x))} \times y_{peak}$$

*Table 6.* Detailed description of degradation functions used to create our dataset.

| Degradation group | Degradation type | Description | Prompt example (inverse) |
|---|---|---|---|
| EQ | X-band EQ | Apply 8 to 12 band parametric EQ with −6 to +6 range for each band. | Correct the unnatural frequency emphasis. |
| | Microphone transfer function | Convolve the audio with one of 20 phone microphone transfer functions. | Reduce the coloration added by the microphone. |
| | Brightness | Reduce brightness using a high-shelf filter at 6 kHz by 6–15 dB. | Give the mix more shine and sparkle. |
| | Darkness | Increase perceived brightness with a high-shelf filter at 6 kHz by 6–15 dB. | Make the tone fuller and less sharp. |
| | Airiness | Reduce airiness via a high-shelf filter at 10 kHz by 10–20 dB. | Add more air and openness to the sound. |
| | Boominess | Reduce boominess with a low-shelf filter at 120 Hz by 10–20 dB. | Give the audio more roar and low-end power. |
| | Clarity | Degrade clarity using a Butterworth low-pass filter (order 3–5) with cutoff at 2 kHz. | Increase the clarity of this song by emphasizing treble frequencies. |
| | Muddiness | Increase muddiness with a 2nd-order Chebyshev Type II bandpass (200–500 Hz) by 6–15 dB. | Make the mix sound less boxy and congested. |
| | Warmth | Reduce warmth with a low-shelf filter at 400 Hz by 6–20 dB. | Make the sound warmer and more inviting. |
| | Vocals | Attenuate vocal-range frequencies using a 2nd-order Chebyshev Type II bandpass (350–3500 Hz) by 6–20 dB. | Make the vocals stand out more. |
| Dynamics | Compression | Apply a feedforward compressor with attack 3–80 ms, release 80–250 ms, threshold −45 to −38 dB, ratio 6–45, and make-up gain 16–25 dB. | Let the audio breathe more and improve the dynamics. |
| | Punch | Apply a feedforward transient shaper with attack 3 ms, release 150 ms, adaptive threshold, and reduction of 8–15 dB. | Add more impact and dynamic punch to the sound. |
| Reverb | Small room | Convolve with Pyroomacoustics simulated IR: room size (7–15, 8–18, 4–14) m, absorption coefficient 0.05–0.30. | Clean this off any echoes! |
| | Big room | Convolve with Pyroomacoustics IR: room size (4–8, 4–7, 2.5–3.5) m, 1–2 absorptive walls, frequency-dependent absorption. | Can you remove the excess reverb in this audio, please? |
| | Mixed material room | Convolve with Pyroomacoustics IR: room size (3–7, 3–9, 2.5–4) m, absorption coefficient 0.05–0.30. | Remove excess reverb and make it sound cleaner. |
| | Real RIR | Apply one of twelve real impulse responses from the openAIR library. | Please, reduce the strong echo in this song. |
| Amplitude | Clipping | Modify the audio level to a maximum amplitude of {2,3,5} and apply clipping. | Reduce the clipping and reconstruct the lost audio, please. |
| | Volume | Adjust the audio gain to a maximum amplitude of {0.001, 0.003, 0.01, 0.05}. | Enhance the loudness without distorting the signal. |
| Stereo | Stereo | Combine the left and right channels to erase the spatial image. | Add depth and separation between left and right. |

| Group (weight) | Option | Probability / Weight |
|---|---|---|
| EQ (0.4) | xband | 7.0 |
| | mic | 5.0 |
| | bright | 3.0 |
| | dark | 3.0 |
| | airy | 2.0 |
| | boom | 2.0 |
| | clarity | 3.0 |
| | mud | 3.0 |
| | warm | 3.0 |
| | vocal | 4.0 |
| Dynamics (0.125) | comp | 2.5 |
| | punch | 1.0 |
| Reverb (0.225) | small | 0.15 |
| | big | 0.15 |
| | mix | 0.30 |
| | real | 0.40 |
| Amplitude (0.125) | clip | 3.0 |
| | volume | 1.0 |
| Stereo (0.125) | stereo | 1.0 |

*Table 7.* Degradation groups with assigned probabilities and option weights.

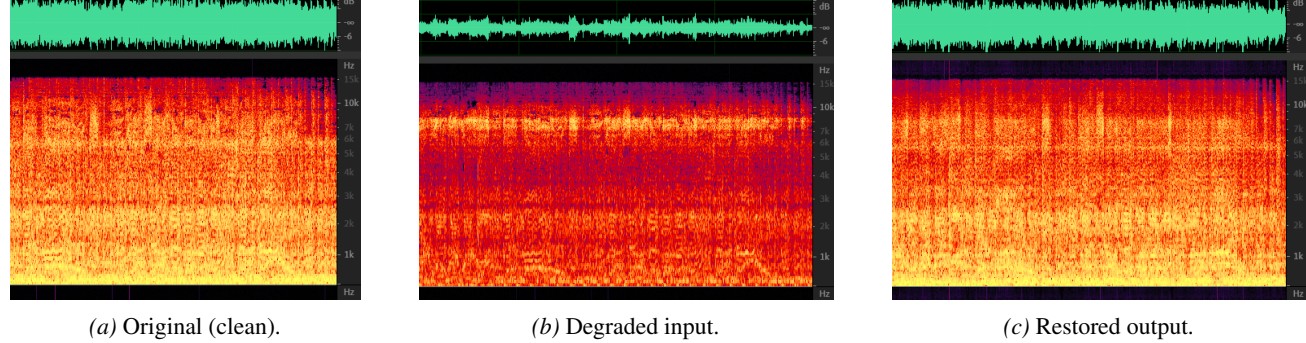

*(a)* Original (clean).                *(b)* Degraded input.                *(c)* Restored output.

*Figure 4.* Original vs. degraded (via convolution with a phone microphone transfer function) and *SonicMaster*-restored spectrograms; restoration suppresses the microphone's coloration.

## A.5. Evaluation metrics details

To evaluate SonicMaster's ability to deal with each of the 19 proposed degradations, we use a set of evaluation metrics as follows in this section. For all the metrics, except for X-band EQ, microphone transfer function, and all reverb options, we report absolute errors, i.e., the absolute value of difference of ground truth (GT) and inferred sample metric values:

$$AbsError_{metric} = \left| metric_{ground\_truth} - metric_{inferred} \right|.$$

**EQ:** The effect of all the EQ options, except for "xband" and "mic" is evaluated through absolute error of spectral energy ratio of two signals – the ground truth reference and the inferred signals. Spectral energy ratio ($Spectral\_ER$) is computed as:

$$Spectral\_ER = \frac{E_{band}}{E_{total}},$$

where $E_{total}$ is the total energy of the signal, and $E_{band}$ is the signal's energy in a spectral band given by the following boundaries $B$:

$$B = \begin{cases} (20, 150), & \text{if "boom"} \\ (20, 400), & \text{if "warm"} \\ (200, 500), & \text{if "mud"} \\ (350, 3500), & \text{if "vocal"} \\ (4000, f_s/2), & \text{if "clarity"} \\ (6000, f_s/2), & \text{if "bright"} \\ (6000, f_s/2), & \text{if "dark"} \\ (10000, f_s/2), & \text{if "airy"} \end{cases}$$

where $f_s$ stands for sampling rate.

The remaining two EQ functions of "xband" and "mic" are evaluated through a cosine distance of spectral balance of the ground truth reference and inferred signal. Spectral balance ($SB$) is calculated as a normalised energy profile in 9 pre-defined frequency bands:

$$SB = \frac{[E_1, E_2, E_3, E_4, E_5, E_6, E_7, E_8, E_9]}{sum[E_1, E_2, E_3, E_4, E_5, E_6, E_7, E_8, E_9]}.$$

The bands are given as:

$$B_{balance} = \begin{cases} (20, 60), & \text{if index} = 1 \\ (60, 250), & \text{if index} = 2 \\ (250, 500), & \text{if index} = 3 \\ (500, 2000), & \text{if index} = 4 \\ (2000, 4000), & \text{if index} = 5 \\ (4000, 6000), & \text{if index} = 6 \\ (6000, 10000), & \text{if index} = 7 \\ (10000, 16000), & \text{if index} = 8 \\ (16000, 20000), & \text{if index} = 9 \end{cases}$$

The reported cosine distance is then gained as:

$$cosine\_distance = 1 - cos(SB_{ground\_truth}, SB_{inferred}).$$

**Amplitude:** Clipping correction is evaluated through spectral flatness using the LIBROSA.FEATURE.SPECTRAL_FLATNESS library function, which takes in a power spectrogram gained through STFT with n_fft=2048 and hop length = 512. The final metric for clipping is the absolute error of spectral flatness (GT vs inferred sample).

Volume is evaluated as the absolute error of the Root-Mean-Square (RMS) value.

**Dynamics:** Compression is evaluated as the standard deviation of the dynamic range ($STD\_DR$), given as:

$$STD\_DR = std(RMS(\mathcal{F}_{H,L})),$$

where $\mathcal{F}_{H,L}$ represents a set of waveform frames with length 2048 and hop length 1024 each. The final metric is the absolute error from the GT.

The "punch" is measured through transient strength by taking the mean value of the transient envelope gained from the LIBROSA.ONEST.ONSET
STRENGTH library function with default parameters.

**Reverb:** We evaluate the effect of dereverberation using modulation spectrum distance ($MSD$).

First, we get a set of temporal envelopes $E_x$ from input signal $x$:

$$E_x^{(k)}(m) = \left| \text{STFT}\{x\}(k, m) \right|,$$

where $k$ indexes frequency bins and $m$ indexes time frames. Modulation spectrum $S_x^{(k)}(b)$ is then calculated using demeaned temporal envelopes:

$$S_x^{(k)}(b) = \left| \text{FFT}_m \left( E_x^{(k)}(m) - \frac{1}{M} \sum_{m'=0}^{M-1} E_x^{(k)}(m') \right) \right|_b, \qquad b = 0, \ldots, B-1.$$

where $b$ represents modulation bins.

Modulation spectra from all frequency bands are then stacked into a single vector:

$$\mathbf{s}_x = \text{vec}\left( S_x^{(k)}(b) \right),$$

and $\ell_2$ normalized:

$$\hat{\mathbf{s}}_x = \frac{\mathbf{s}_x}{\|\mathbf{s}_x\|_2 + \varepsilon}.$$

The MSD between two signals, in our case the GT reference $x_{GT}$ and relevant inferred sample $x_{infer}$, is given as Euclidean distance:

$$MSD(x_{GT}, x_{infer}) = \left\| \hat{\mathbf{s}}_{x_{GT}} - \hat{\mathbf{s}}_{x_{infer}} \right\|_2.$$

In code, this is realized with following parameters as:

```
import numpy as np
from scipy.spatial.distance import euclidean
from scipy.signal import stft

def modulation_spectrum_distance(x1, x2, fs=44100,
    n_fft=1024, hop_length=512, n_mod_bins=20):
```

```
def get_modulation_spectrum(x):
    f, t, Zxx = stft(x, fs=fs, nperseg=n_fft, noverlap=n_fft - hop_length)
    mag = np.abs(Zxx)

    mod_spec = []
    for band in mag:
        envelope = band - np.mean(band)
        spectrum = np.abs(np.fft.fft(envelope))[:n_mod_bins]
        mod_spec.append(spectrum)

    mod_spec = np.array(mod_spec)
    mod_spec /= np.linalg.norm(mod_spec) + 1e-10
    return mod_spec.flatten()

mod1 = get_modulation_spectrum(x1)
mod2 = get_modulation_spectrum(x2)

return euclidean(mod1, mod2)
```

**Stereo:** We measure the level of stereoness using stereo energy ratio ($Stereo\_ER$), computed as:

$$Stereo\_ER = \frac{RMS(\frac{L-R}{2})}{RMS(\frac{L+R}{2}) + 10^{-10}} \tag{11}$$

We report the absolute error of this metric.

### A.6. Spectrogram examples

We visualize time–frequency structure in spectrograms to provide qualitative evidence of restoration behavior. Each figure shows the clean reference, the degraded input (e.g., reverberation-induced smearing or clipping distortion), and the output of SonicMaster. Figures 4, 5, 6, 7, 8, and 9 compare clean, degraded, and restored spectrograms across selected scenarios (reverb, clipping, microphone transfer function, and clarity EQ).

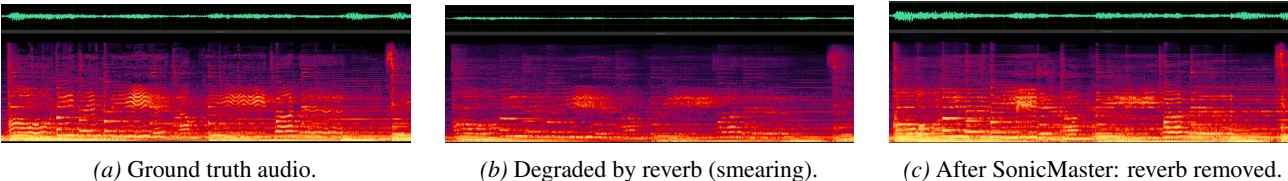

*(a)* Ground truth audio.          *(b)* Degraded by reverb (smearing).          *(c)* After SonicMaster: reverb removed.

*Figure 5.* Comparison of spectrograms: (a) ground truth, (b) degraded with reverb, and (c) the output of SonicMaster where smearing is removed. Prompt: "Please, reduce the strong echo in this song."

### A.7. Subjective evaluation – first listening test design example

We provide an example of the first listening test design, which was carried out using the Psytoolkit platform (Stoet, 2010; 2017). Figure 10 shows the user interface with two audio samples (the input and the modified output), the instruction, the guiding prompt of the desired change, and the questions for the 5 aspects rated on the 7-point Likert Scale. We copy the briefing text and an example of the guiding text for each example below:

**Briefing:**

Hey! Thank you for participating in this listening test. :-)

After the initial survey questions about your background, you will encounter some 40 questions with audio samples (which should take you some 20-30 minutes). Each question will have 2 samples that form a pair. The first (top) sample represents

the input, the second (bottom) sample represents the output after applying a certain transformation described by the text prompt, e.g., "Make this audio brighter" should make the second sample sound brighter compared to the first sample; "Remove the echo!" should make the second audio have less of echoey reverb present than the first audio.

Please use headphones! You do not need to listen to the whole audio samples, just listen enough to make a conscious judgement.

**Sample example:**

Instructions: There are two audio samples above. The second one is a modified version of the first one. This modification was done according to the Prompt. Please listen to both the audio clips, read the Prompt, and answer the questions below.

Prompt: Lift the treble for a more open tone.

1) To what extent does the second audio sample represent the intended change given by the Prompt?

2) How good is the audio quality of the first sample?

3) How good is the audio quality of the second sample?

4) How fluent and coherent is the second sample?

5) Do you prefer the second sample over the first sample?

### A.8. Subjective evaluation – second listening test

The comparative evaluation against existing baselines demonstrates *SonicMaster*'s superior performance across both reverb and EQ degradation categories (Figure 11). For reverb artifacts, participants overwhelmingly preferred *SonicMaster* over Mel2Mel + Diffwave (Kandpal et al., 2022), selecting our method in 191 out of 200 total comparisons (10 samples × 20 participants), with Mel2Mel + Diffwave chosen only twice. In the EQ category, *SonicMaster* achieved similarly strong results with 180 out of 200 preferences, while Mel2Mel + Diffwave received 13 votes, Text2FX (Chu et al., 2025) garnered 4 votes and Text2FX-directional generated 3. These results show *SonicMaster*'s effectiveness in addressing both spatial acoustic degradations and spectral imbalances.

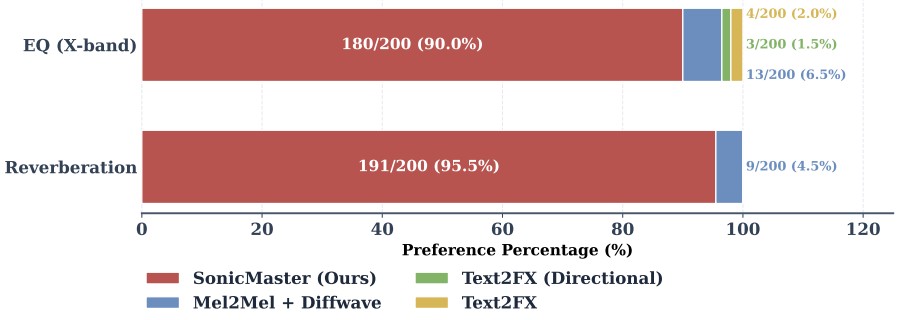

*Figure 11.* Comparative Listening Study Results ($N = 20$ participants × 10 samples per category).

### A.9. Ablation on ODE solvers, model size, and conditioning

We evaluated Euler solvers with 1, 10 (baseline), and 100 steps, plus a 10-step 4th order Runge–Kutta (Dormand & Prince, 1980) solver. Tables 8, 9, and 10 outline the results and highlight the trade-off across degradation categories. Euler-1 matches the baseline overall but is weaker on Boom, Microphone, Clip, all Reverb subtasks, and shows higher KL. Euler-100 boosts Reverb and Punch yet lowers every EQ score versus the 1-/10-step runs. Runge–Kutta-10 equals Euler-10 on most metrics and tops Clip, but its inference is significantly slower.

We further performed a scaling analysis of the *SonicMaster* model. The results in Tables 8, 9, 10, show that $SonicMaster_{Small}$ performs comparably with $SonicMaster_{Large}$ in all metrics, but slightly worse in Reverb, Clip, and Stereo. The medium variant, $SonicMaster_{Medium}$ (4MM-DiT/12DiT), performs slightly better than the small model $SonicMaster_{Small}$ overall. It also performs comparably to the large model $SonicMaster_{Large}$, outperforming it in Boom, or Compression, but still lacking

| Model (MMDiT/DiT) | Clarity | Boom | Airy | Bright | Dark | Muddy | Warm | Vocals | Microphone | X-band |
|---|---|---|---|---|---|---|---|---|---|---|
| **Snippet degraded input** | | | | | | | | | | |
| *SonicMaster*$_{Large}$ (6/18) | 0.0114 | 0.0834 | **0.0019** | 0.0059 | **0.0058** | 0.0388 | **0.0617** | 0.0576 | 0.0088 | **0.0358** |
| -w Euler 1 Step | **0.0100** | 0.1146 | 0.0019 | 0.0059 | 0.0061 | 0.0425 | 0.0668 | **0.0498** | 0.0141 | 0.0384 |
| -w Euler 100 Steps | 0.0136 | 0.1540 | 0.0033 | 0.0100 | 0.0091 | 0.0540 | 0.0915 | 0.0749 | 0.0162 | 0.0444 |
| -w Runge-Kutta 10 Steps | 0.0120 | **0.0810** | 0.0019 | **0.0058** | 0.0058 | 0.0402 | 0.0630 | 0.0590 | **0.0083** | 0.0374 |
| *SonicMaster*$_{Small}$ (2/6) | **0.0100** | 0.0819 | 0.0020 | 0.0064 | 0.0060 | 0.0477 | **0.0590** | 0.0630 | 0.0122 | 0.0408 |
| *SonicMaster*$_{Medium}$ (4/12) | 0.0105 | **0.0698** | 0.0021 | 0.0067 | 0.0061 | 0.0400 | 0.0592 | 0.0602 | 0.0091 | 0.0383 |
| *SonicMaster*$_{Medium}$ (6/6) | 0.0225 | 0.2766 | 0.0020 | 0.0067 | **0.0056** | 0.1718 | 0.1737 | 0.2417 | 0.0462 | 0.0762 |
| *SonicMaster*$_{Large}$ (6/18) | 0.0114 | 0.0834 | **0.0019** | 0.0059 | 0.0058 | **0.0388** | 0.0617 | **0.0576** | **0.0088** | **0.0358** |
| *SonicMaster*$_{Large}$ (6/18) | **0.0114** | **0.0834** | **0.0019** | **0.0059** | **0.0058** | **0.0388** | **0.0617** | 0.0576 | **0.0088** | **0.0358** |
| -w No Text Condition | 0.0130 | 0.1432 | 0.0032 | 0.0101 | 0.0086 | 0.0448 | 0.0841 | 0.0668 | 0.0154 | 0.0424 |
| -w Shuffled Prompts | 0.0187 | 0.2075 | 0.0077 | 0.0132 | 0.0362 | 0.0981 | 0.1648 | 0.1043 | 0.0424 | 0.0998 |
| *SonicMaster*$_{Large}$ (6/18) | 0.0114 | 0.0834 | **0.0019** | **0.0059** | 0.0058 | 0.0388 | 0.0617 | 0.0576 | 0.0088 | 0.0358 |
| -w 5s Audio Cond. | 0.0111 | 0.0716 | 0.0021 | 0.0061 | 0.0058 | 0.0386 | 0.0605 | 0.0628 | 0.0124 | 0.0387 |
| -w 15s Audio Cond. | 0.0117 | 0.0750 | 0.0020 | 0.0064 | 0.0063 | **0.0320** | **0.0552** | 0.0525 | **0.0079** | 0.0398 |
| -w/o Audio Cond. (basic) | **0.0099** | **0.0658** | 0.0021 | 0.0064 | **0.0056** | 0.0352 | 0.0595 | 0.0746 | 0.0097 | 0.0434 |
| -w Cond. During Infer | 0.0115 | 0.0840 | 0.0019 | 0.0060 | 0.0058 | 0.0389 | 0.0610 | 0.0572 | 0.0088 | **0.0355** |

*Table 8.* EQ Objective evaluation (average absolute error) – the lower, the better.

behind in Clip. *SonicMaster*$_{Medium}$ (6MM-DiT/6DiT) performs the worst out of all variants across all metrics, suggesting a non-optimal ratio of MM-DiT to DiT blocks.

Regarding the audio condition and its duration, we evaluated *SonicMaster*$_{Large}$ with three different conditioning lengths (5s, 10s, 15s). The performance across configurations was found to be comparable (Tables 8, 9, 10). For our default model version, we chose the 10-second setting as it balances computational efficiency with temporal overlap for long-form processing. The variant that uses audio condition through the pooling layers during inference scored comparably to the default setup, however, we can observe improvement in Clip and Volume (Table 9). The model trained without audio conditioning performs similarly across the board, scoring the best in Boom (0.0658, Table 8), but shows a clear drop in Clip performance (2.055 vs 1.506, see Table 9), which highlights the importance of this condition for this reconstruction task.

Inference without text prompts maintains comparable FAD, SSIM, and PQ but shows degraded KL divergence (0.917 vs. 0.696). Critical drops occur in Clip restoration (2.812 vs. 1.506) and Stereo processing (0.1416 vs. 0.1058), with elevated EQ errors. To further assess the text controllability, we shuffled the prompts inside the test set and ran inference. Results (Tables 8, 9, 10 show worse performance than when no prompt was given (KL 2.014, Clip 2.874), but still show large improvement over the degraded input. This confirms text conditioning enables targeted restoration rather than generic improvements.

### A.10. Limitations and future work

In this section, we provide a whole list of limitations of the current work and sketch out potential future work.

**Cultural bias:** Even though our dataset consists of 10 genre groups, each represented by many genre tags, the vast majority of the music comes from Western culture. Thus, there is a bias in the data and the model functionality. Future studies can expand this by focusing on music from other regions of the world.

**Dataset:** When creating the SonicMaster dataset, we tried to emulate real-world conditions by: 1) specifically applying real-world room impulse responses from the openAIR library (Howard & Angus, n.d.) and phone microphone transfer functions from the POLIPHONE dataset (Salvi et al., 2025); 2) implementing random ranges for a large number of parameters, covering all degradation functions except for the amplitude and stereo functions, which contained a limited number of options.

We note that further improvements to the data can be achieved by: a) expanding the dataset to cover more functions, artifacts, genres, instruments, and more; b) collecting a real-world based dataset to better capture and represent any (and all) potential artifacts present in the real world.

| Model (MMDiT/DiT) | Reverb | | | | Dynamics | | Amplitude | | Stereo |
|---|---|---|---|---|---|---|---|---|---|
| | Small | Big | Mix | Real | Compressor | Punch | Clip | Volume | |
| **Snippet degraded input** | | | | | | | | | |
| *SonicMaster*Large (6/18) | 0.3663 | 0.3726 | 0.3935 | 0.3109 | 0.0193 | 0.0871 | 1.506 | 0.0468 | **0.1058** |
| -w Euler 1 step | 0.4215 | 0.4378 | 0.4599 | 0.3459 | **0.0124** | 0.0906 | 2.171 | **0.0461** | 0.1261 |
| -w Euler 100 Steps | 0.3716 | 0.3754 | 0.3997 | 0.3255 | 0.0158 | **0.0672** | 2.753 | 0.0491 | 0.1497 |
| -w Runge-Kutta 10 Steps | **0.3647** | **0.3684** | **0.3921** | **0.3087** | 0.0210 | 0.0858 | **1.422** | 0.0481 | 0.1059 |
| *SonicMaster*Small (2/6) | 0.3812 | 0.3826 | 0.4050 | 0.3277 | 0.0172 | 0.0859 | 2.363 | 0.0457 | 0.1536 |
| *SonicMaster*Medium (4/12) | 0.3683 | **0.3700** | **0.3934** | 0.3138 | **0.0147** | 0.0891 | 2.455 | **0.0409** | **0.1028** |
| *SonicMaster*Medium (6/6) | 0.3952 | 0.3916 | 0.4422 | 0.4255 | 0.0366 | **0.0833** | 2.905 | 0.1228 | 0.4180 |
| *SonicMaster*Large (6/18) | **0.3663** | 0.3726 | 0.3935 | **0.3109** | 0.0193 | 0.0871 | **1.506** | 0.0468 | 0.1058 |
| *SonicMaster*Large (6/18) | **0.3663** | **0.3726** | **0.3935** | **0.3109** | 0.0193 | 0.0871 | **1.506** | **0.0468** | **0.1058** |
| -w No Text Condition | 0.3732 | 0.3805 | 0.4012 | 0.3264 | **0.0157** | **0.0730** | 2.812 | 0.0465 | 0.1416 |
| -w Shuffled Prompts | 0.4161 | 0.4236 | 0.4538 | 0.3903 | 0.0225 | 0.0895 | 2.874 | 0.0895 | 0.3213 |
| *SonicMaster*Large (6/18) | 0.3663 | 0.3726 | 0.3935 | 0.3109 | 0.0193 | 0.0871 | 1.506 | 0.0468 | 0.1058 |
| -w 5s Audio Cond. | 0.3717 | **0.3658** | 0.3919 | 0.3079 | 0.0164 | 0.0893 | 1.779 | 0.0430 | **0.0918** |
| -w 15s Audio Cond. | 0.3676 | 0.3682 | 0.3901 | 0.3093 | 0.0172 | 0.0895 | 1.633 | 0.0485 | 0.1008 |
| -w/o Audio Cond. During Training | **0.3620** | 0.3682 | **0.3888** | 0.3067 | **0.0146** | 0.0850 | 2.055 | 0.0455 | 0.1015 |
| -w Cond. During inference | 0.3664 | 0.3724 | 0.3934 | 0.3112 | 0.0172 | 0.0870 | **1.455** | **0.0412** | 0.1060 |

*Table 9.* Objective evaluation: Reverb, Dynamics, Amplitude, and Stereo. Clip values are multiplied by 1000.

**Language:** Our target audience were amateur musicians, which we assume do not use too much of music production/mastering jargon. We aimed to develop a dataset with sufficiently rich language by covering each of the functions with 7 to 10 different prompts to cover for the needs of these users. However, the ways in which people can describe music are virtually limitless. A robust text-encoder, such as the used FLAN-T5 can somewhat account for this, given that synonyms and semantically similar sentences are typically located close to one another in the latent space. However, this may not hold for highly specialised jargon, such as that used in music production and mastering, where word meanings may differ slightly or significantly from their conventional usage. For example, "cold" and "bright" may both refer to music with rich high-frequency content, yet they evoke very different real-world associations, where brightness may be linked to sunlight and coldness to the opposite. Alongside this sketched expansion of language, the aspect of language semantics should also be considered and studied. The instructions used to control such an automated music production system could be, for example, subjected to feasibility tests.

**Mastering:** Our aim was to develop a model for all-in-one music restoration and mastering. The scope of "mastering", however, can be broader than what was presented. Mastering not only deals with polishing and optimization of single recordings, but also and mainly with coherence across whole albums, format-specific preparation (e.g., vinyl mastering), and more. This presents an avenue for future expansion. Nonetheless, with the current vision of helping amateur musicians, a focus on single recordings feels sufficient, as that is the most expected problem they might want to solve.

**Functionality:** While SonicMaster was trained with 19 common degradations from 5 distinct categories, the list is not exhaustive. Future work can expand this by introducing more functions, that could be related to mixing, such as panning for various instruments present, which would be a very challenging task for a single-track recording.

**Architecture and design:** The current bottleneck of the model seems to be the audio latent space, manifesting as occasional artifacts in the music, especially robotic voice, or slight loses of intelligibility in music with rich content. Further experiments are required to verify this and propose better solutions. Furthermore, the current inference design of chaining segments is not optimal, which sometimes manifests as inconsistent volume over the track's duration. Other approaches assuring more consistency over time should be considered in the future.

### A.11. Prompts for each degradation type

Prompt instructions for each degradation type are grouped by audio attribute in Table 11; for example, entries for Xband, microphone coloration, clarity, brightness, darkness, airiness, boominess, warmth, muddiness, vocals, compression, punch,

| Model | Single deg. | | | | Double+triple deg. | | | | All | | | |
|---|---|---|---|---|---|---|---|---|---|---|---|---|
| | FAD ↓ | KL ↓ | SSIM ↑ | PQ ↑ | FAD ↓ | KL ↓ | SSIM ↑ | PQ ↑ | FAD ↓ | KL ↓ | SSIM ↑ | PQ ↑ |
| **Snippet degraded input** | | | | | | | | | | | | |
| *SonicMaster$_{Large}$* (6/18) | **0.069** | **0.696** | **0.624** | 7.743 | **0.082** | **1.145** | **0.589** | **7.654** | **0.073** | **0.888** | **0.609** | **7.705** |
| -w Euler 1 step | 0.076 | 0.922 | 0.615 | 7.684 | 0.117 | 1.789 | 0.567 | 7.520 | 0.090 | 1.294 | 0.594 | 7.614 |
| -w Euler 100 Steps | 0.069 | 0.920 | 0.620 | **7.764** | 0.087 | 1.521 | 0.585 | 7.621 | 0.076 | 1.178 | 0.605 | 7.703 |
| -w Runge-Kutta 10 Steps | 0.070 | 0.701 | 0.624 | 7.740 | 0.084 | 1.171 | 0.588 | 7.642 | 0.074 | 0.902 | 0.608 | 7.698 |
| *SonicMaster$_{Small}$* (2/6) | 0.071 | 0.726 | 0.623 | 7.716 | 0.088 | 1.215 | 0.586 | 7.609 | 0.077 | 0.935 | 0.607 | 7.670 |
| *SonicMaster$_{Medium}$* (4/12) | 0.070 | 0.709 | 0.624 | 7.740 | 0.084 | 1.187 | 0.589 | 7.649 | 0.075 | 0.914 | 0.609 | 7.701 |
| *SonicMaster$_{Medium}$* (6/6) | 0.086 | 1.893 | 0.603 | 7.571 | 0.154 | 3.241 | 0.555 | 7.231 | 0.110 | 2.470 | 0.583 | 7.426 |
| *SonicMaster$_{Large}$* (6/18) | **0.069** | **0.696** | **0.624** | **7.743** | **0.082** | **1.145** | **0.589** | **7.654** | **0.073** | **0.888** | **0.609** | **7.705** |
| *SonicMaster$_{Large}$* (6/18) | **0.069** | **0.696** | **0.624** | 7.743 | **0.082** | **1.145** | **0.589** | **7.654** | **0.073** | **0.888** | **0.609** | **7.705** |
| -w No Text Condition | 0.069 | 0.917 | 0.621 | **7.772** | 0.088 | 1.484 | 0.586 | 7.643 | 0.074 | 1.160 | 0.606 | 7.716 |
| -w Shuffled Prompts | 0.081 | 2.014 | 0.598 | 7.610 | 0.131 | 3.249 | 0.558 | 7.283 | 0.098 | 2.543 | 0.581 | 7.470 |
| *SonicMaster$_{Large}$* (6/18) | **0.069** | 0.696 | 0.624 | **7.743** | **0.082** | 1.145 | 0.589 | **7.654** | **0.073** | 0.888 | 0.609 | **7.705** |
| -w 5s Audio Cond. | 0.070 | 0.703 | 0.624 | 7.733 | 0.083 | 1.175 | 0.588 | 7.637 | 0.075 | 0.905 | 0.609 | 7.692 |
| -w 15s Audio Cond. | 0.069 | 0.694 | 0.623 | 7.742 | 0.083 | 1.161 | 0.588 | 7.650 | 0.073 | 0.894 | 0.608 | 7.702 |
| -w/o Audio Cond. During Training | 0.069 | **0.691** | 0.625 | 7.741 | 0.082 | 1.146 | **0.590** | 7.645 | 0.073 | 0.886 | **0.610** | 7.700 |
| -w Cond. During Inference | 0.069 | 0.693 | **0.625** | 7.742 | 0.082 | **1.141** | 0.589 | 7.653 | 0.073 | **0.885** | 0.609 | 7.704 |

*Table 10.* Objective evaluation: FAD, KL, SSIM, and PQ. For readability, KL values were multiplied by 1000.

reverb, volume, clipping, and stereo give natural-language commands that steer the restoration model. These instructions act as conditioning signals—e.g., "remove excess reverb and make it sound cleaner," "raise the level of the vocals," or "make this sound brighter"—so that the generative restoration trajectory emphasizes or suppresses specific signal characteristics.

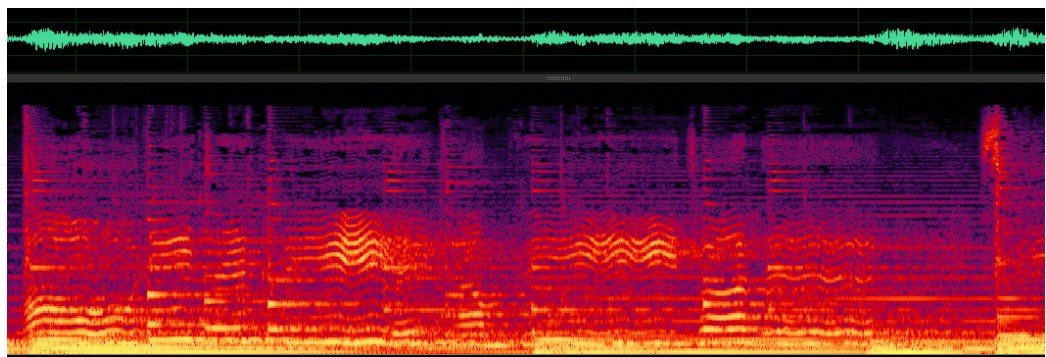

*(a)* Original (clean).

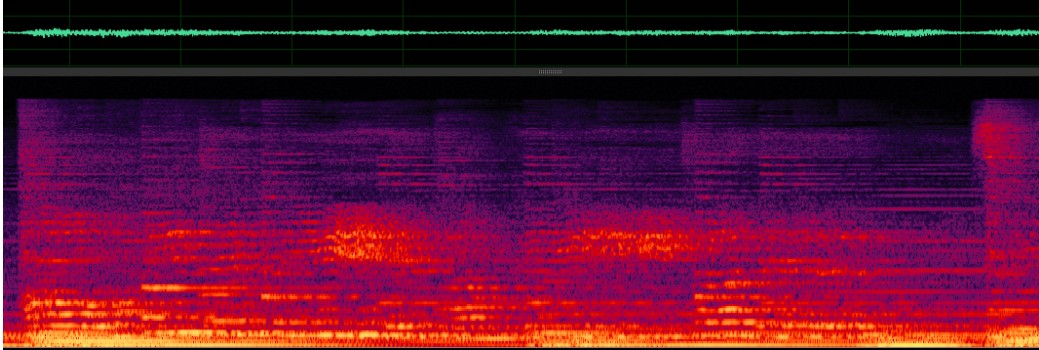

*(b)* Degraded input.

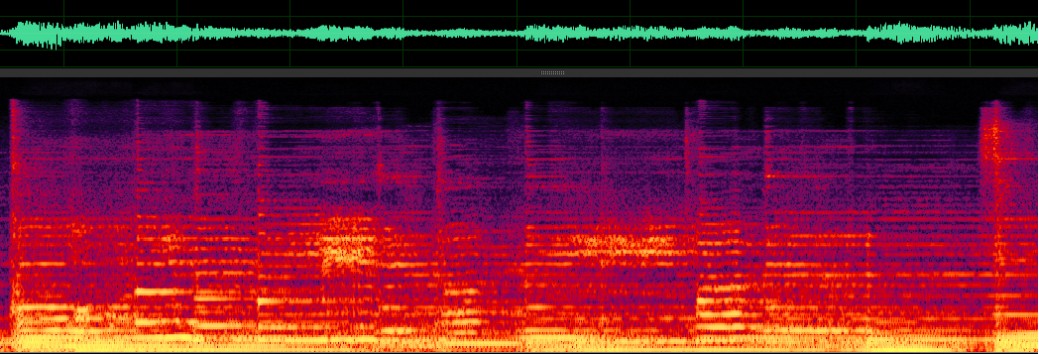

*(c)* Restored output.

*Figure 6.* Effect of reverberation (example from the main text in larger size): top panel shows the original audio sample, middle panel shows audio convolved with a Pyroomacoustics simulated impulse response, and bottom panel shows the dereverberated result with echoes cleaned.

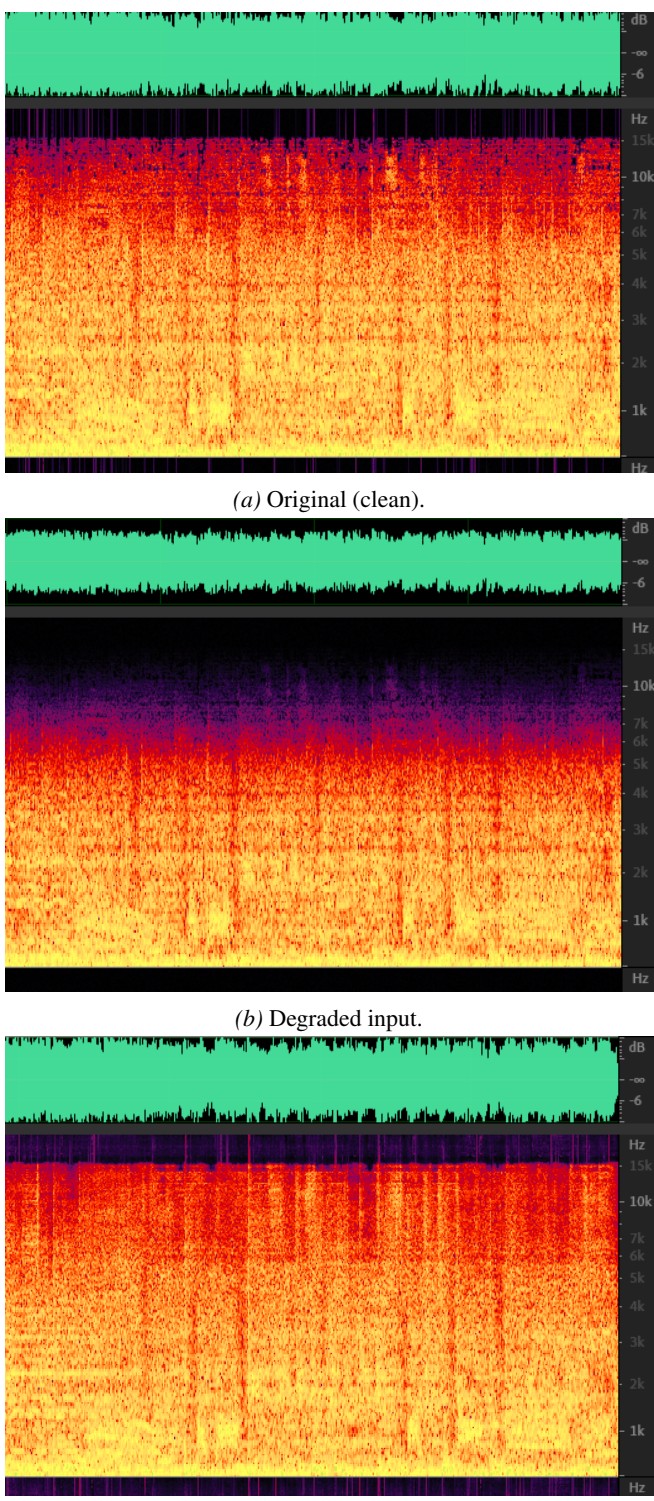

*(a)* Original (clean).

*(b)* Degraded input.

*(c)* Restored output.

*Figure 7.* Effect of clarity degradation and restoration on spectrograms. The treble frequencies are supressed in the degraded input sample, and then restored with SonicMaster. Prompt: "Make the audio clearer and more intelligible."

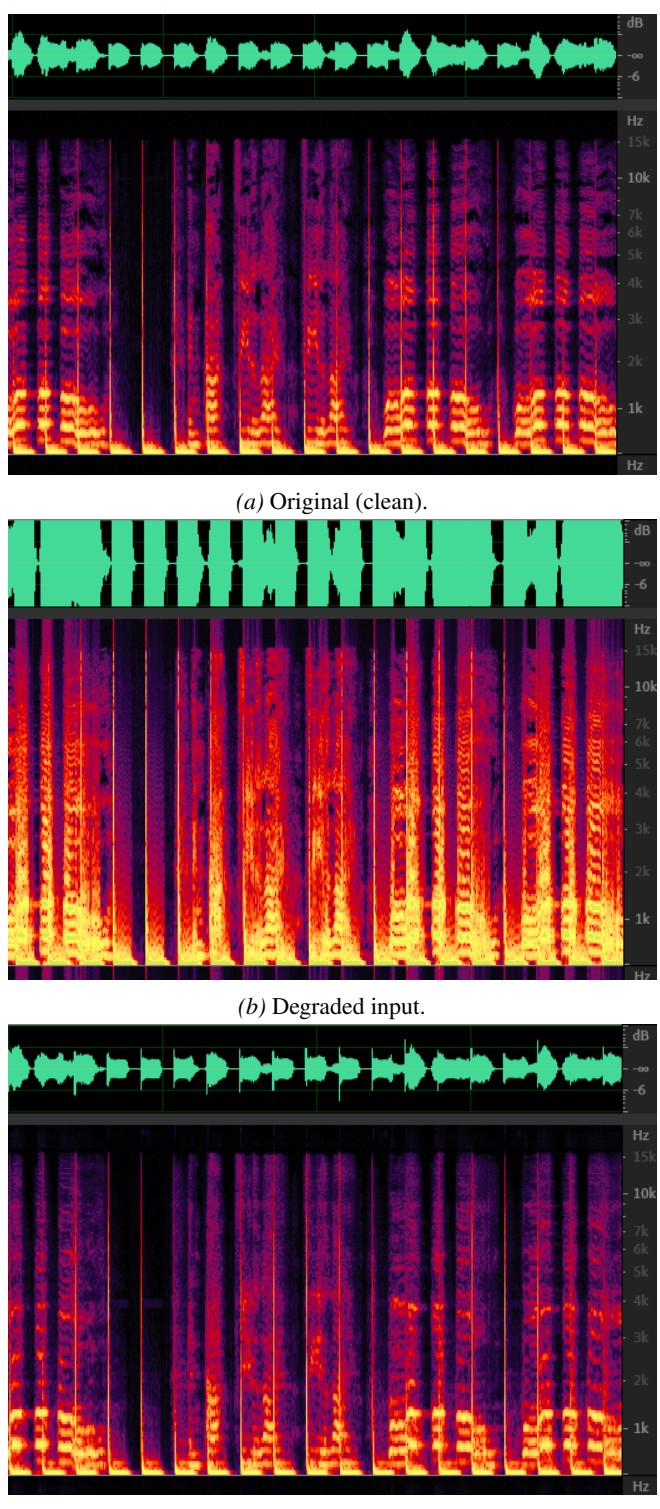

*(a)* Original (clean).

*(b)* Degraded input.

*(c)* Restored output.

*Figure 8.* Effect of clipping degradation and related restoration. Drum hits clip in the degraded audio, showing as wideband spectral peaks, but are restored in the SonicMaster's output without distortion. Prompt: "Clean up the harshness in the signal."

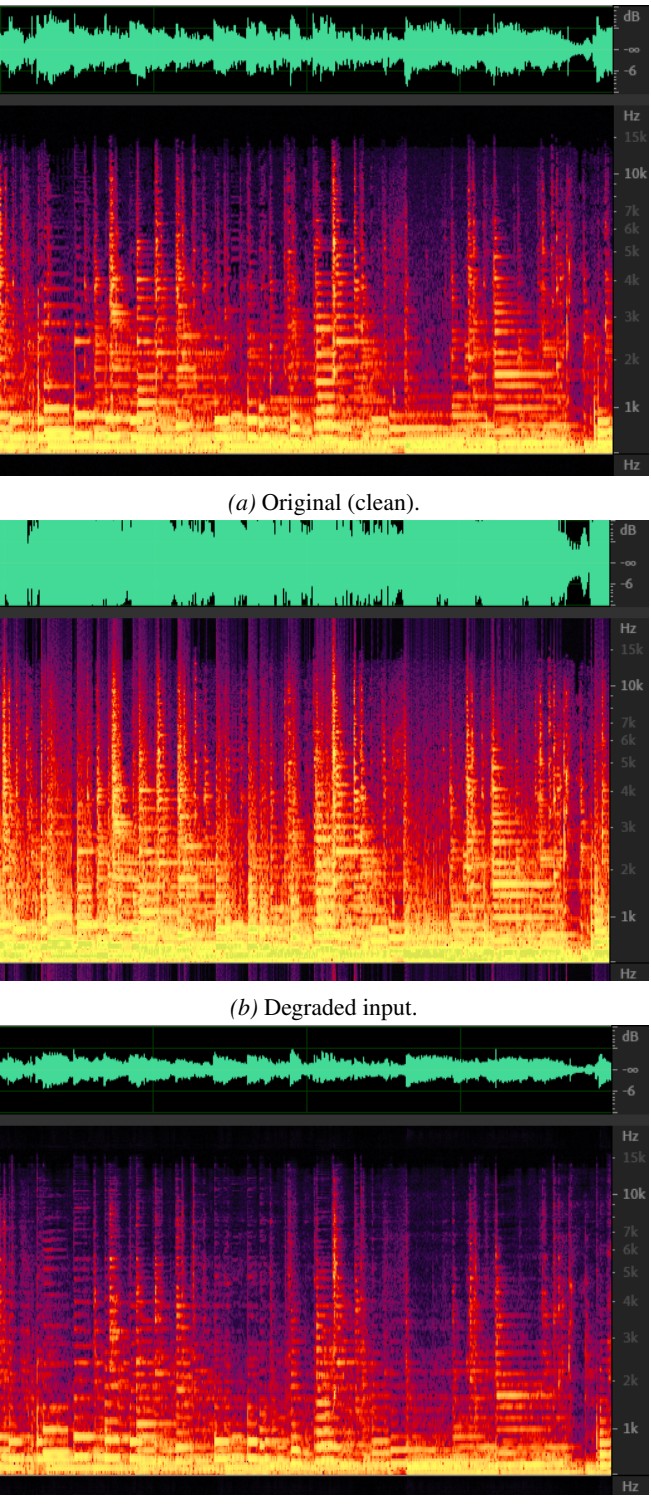

*(a)* Original (clean).

*(b)* Degraded input.

*(c)* Restored output.

*Figure 9.* Another example of the effect of clipping and its restoration. The degraded input shows signs of distortion with visible increase in wideband spectral content at the parts of waveform clipping. This distortion is suppressed by SonicMaster. Prompt: "Clean up the noisiness in the audio."

▶  0:00 / 4:18

▶  0:00 / 4:29

**Instructions:**

There are two audio samples above. The second one is a modified version of the first one. This modification was done according to the **Prompt**. Please listen to both the audio clips, read the **Prompt**, and answer the questions below.

**Prompt**: Lift the treble for a more open tone.

To what extent does the **second audio sample** represent the intended change given by the **Prompt**?

| 1 Not at all | 2 | 3 | 4 Neutral | 5 | 6 | 7 Very much |
|---|---|---|---|---|---|---|

How good is the audio quality of the **first** sample?

| 1 Not at all | 2 | 3 | 4 Neutral | 5 | 6 | 7 Very much |
|---|---|---|---|---|---|---|

How good is the audio quality of the **second** sample?

| 1 Not at all | 2 | 3 | 4 Neutral | 5 | 6 | 7 Very much |
|---|---|---|---|---|---|---|

How fluent and coherent is the **second** sample?

| 1 Not at all | 2 | 3 | 4 Neutral | 5 | 6 | 7 Very much |
|---|---|---|---|---|---|---|

Do you prefer the **second** sample **over** the **first** sample?

| 1 Not at all | 2 | 3 | 4 Neutral | 5 | 6 | 7 Very much |
|---|---|---|---|---|---|---|

*Figure 10.* An example from the first listening test showcasing the editing Prompt, the instructions for listeners, the two audio samples, and the 5 categories to rate.

*Table 11.* User instructions grouped by audio attribute.

| Attribute | Example Instructions |
|---|---|
| Xband | Can you please correct the equalization?; Improve the balance in the audio by fixing the chaotic equalizer, please.; Make this sound balanced, please.; Balance the EQ, please.; Balance the tonal spectrum of the audio.; Correct the unnatural frequency emphasis.; Make the EQ curve smoother and more natural.; Even out the EQ.; Adjust the tonal balance for a more pleasing sound. |
| Microphone | This audio was recorded with a phone, can you fix that, please?; Please make this sound better than a phone recording.; Balance the EQ, please.; Improve the balance in this song.; Make the audio sound like it was recorded with a higher-quality microphone.; Reduce the coloration added by the microphone.; Make the tone more neutral and balanced.; Improve the naturalness of the recording.; Remove the harshness or boxiness from the mic coloration. |
| Clarity | Increase the clarity!; Can you please make this song sound more clear?; Increase the clarity of this song by emphasizing treble frequencies.; Make the audio clearer and more intelligible.; Sharpen the overall sound.; Bring more focus and definition to the details.; Make the mix sound less cloudy.; Tighten the articulation in the sound. |
| Brightness | Can you please make this sound brighter?; Increase the brightness!; Make this audio sound brighter by emphasizing the high frequencies.; Add some brightness to the high end.; Make the sound more vivid and lively.; Give the mix more shine and sparkle.; Lift the treble for a more open tone.; Enhance the presence of the upper frequencies. |
| Darkness | Make this sound darker!; Can you reduce the brightness, please?; Make the audio darker by suppressing the higher frequencies.; Bring in more low-mid richness to make the sound darker.; Make the tone fuller and less sharp.; Smooth out the highs with deeper low-end support.; Round out the sound with more body.; Soften the harshness with a warmer tone. |
| Airiness | Make this sound more fresh and airy by emphasizing the high end frequencies.; Make this feel more airy, please.; Increase the perceived airiness, please.; Give this a light sense of spaciousness by amplifying the higher frequencies.; Add more air and openness to the sound.; Make the audio feel more spacious and extended.; Enhance the sense of space in the highs.; Lift the top end for a more open character.; Give the mix a breathier, more open feel. |
| Boominess | Make it boom!; Make this song sound more boomy by amplifying the low end bass frequencies.; Increase the boominess, please!; Give me more bass!; Can you make this more bassy, please?; Give the audio more roar and low-end power.; Make the bass more impactful and solid.; Add weight and depth to the bottom end.; Reinforce the low frequencies for more energy.; Boost the bass presence. |
| Warmth | Can you make this song sound warmer, please?; Increase the warmth, please.; Emphasize the bass and low-mid frequencies to give this a more warm feel.; Make the sound warmer and more inviting.; Add some low-mid warmth to the mix.; Soften the tone with a bit more body.; Give the audio a warm analog feel.; Enhance the warmth for a fuller sound. |
| Muddiness | Can you make this song sound less muddy, please?; Decrease the muddiness!; Reduce the level of muddiness in this audio by lowering the low-mid frequencies.; Clean up the muddiness in the low-mids.; Make the mix sound less boxy and congested.; Improve definition by reducing mud.; Clear up the low-mid buildup.; Make the audio tighter and less murky. |
| Vocals | Raise the level of the vocals, please.; Can you amplify the vocals, please?; Emphasize the vocals by raising the level of the mid frequencies specific for vocals.; Bring the vocals forward in the mix.; Make the voice clearer and more present.; Increase the vocal presence by enhancing the midrange.; Make the vocals stand out more.; Strengthen the vocal clarity and focus. |
| Compression | Increase the dynamic range.; Decompress the audio, please.; Remove the compression, please.; Can you fix the strong compression effect in this audio by expanding the dynamic range?; Restore the dynamics of the audio.; Make the sound less squashed and more open.; Reduce the over-compression for a more natural feel.; Bring back the contrast in volume.; Let the audio breathe more and improve the dynamics. |
| Punch | Give this song a punch!; Make the transients sharper, please.; Increase the punchiness of the song by emphasizing the transients.; Make the audio more punchy and energetic.; Bring back the snap and attack of transients.; Add more impact and dynamic punch to the sound.; Make drums and hits sound more aggressive and tight.; Increase the percussive clarity and definition. |
| Reverb | Can you remove the excess reverb in this audio, please?; Please, dereverb this audio.; Remove the echo!; Please, reduce the strong echo in this song.; Remove the church effect, please.; Clean this off any echoes!; This song has too much reverb present, can you reduce it?; Make the audio sound more dry and direct.; Reduce the roominess or echo.; Remove excess reverb and make it sound cleaner.; Bring the sound closer and more focused.; Tighten the spatial feel of the audio. |
| Volume | The volume is low, make this louder please!; Can you make this sound louder, please?; Increase the amplitude.; Normalize the audio volume.; Make the audio louder and more powerful.; Increase the overall level.; Boost the volume without distorting the signal. |
| Clipping | This audio is clipping, can you please remove it?; Remove the loud hissing in this song?; Remove the clipping.; Reduce the clipping and reconstruct lost audio.; Clean up noisiness.; Make the audio smoother and less distorted.; Reduce the gritty or crushed character.; Fix digital distortion. |
| Stereo | Make it sound spacious!; Can you make this audio stereo, please?; Alter left/right channels to give spatial feel.; Widen the stereo image.; Add depth and separation between left and right.; Enhance the stereo field for immersive sound. |

