# OpenReview forum: "SonicMaster: Towards Controllable All-in-One Music Restoration and Mastering"
_ICML.cc/2026/Conference — ICML 2026 regular_

### Official Review · Reviewer_zJqJ · 2026-03-02

**Soundness:** 3
**Presentation:** 3
**Significance:** 3
**Originality:** 3
**Overall Recommendation:** 4
**Confidence:** 4

**Summary:**

The paper proposes a model for text-based (or fully automated) restoration and possibly mastering of music audio recordings. The model is trained on a set of recordings where specific degradations have been applied, and the model uses a flow matching generative approach. Experiments are carried out in comparison with several baselines across both objective metrics, plus a subjective listening test.

**Compliance With Llm Reviewing Policy:**

Affirmed.

**Final Justification:**

This paper possibly is the first to address this particular problem using a single model, and the proposed method is technically correct and appropriate. Most minor comments have been addressed in the rebuttal, however I had some concerns regarding the use of the term "mastering" throughout the paper, including the title and abstract - which has not been fully addressed in the rebuttal. My general recommendation still remains as a weak accept.

**Key Questions For Authors:**

No specific questions to ask the authors.

**Limitations:**

* As mentioned above, the paper claims to address both music restoration and mastering, however in my view this work does not adequately address mastering topics, which include but also go beyond one single recording and is often applied at an album level. In my view, the paper's title and overall narrative needs to be modified in order to address the scope and focus of this work. Even addressing multiple degradations and processes at a track-level is commendable.
* The paper's reference style is not correct - please use carefully either \cite or \citep when needed depending on the reference and sentence.
* Was ethics approval needed in order to carry out the listening test? If yes, please mention this. If not, please also mention this - this is dependent on the institutional policies.
* The way the listening test was carried out was unclear. Was there a comparison between the proposed model and other models, or was the comparison only on the basis of the input audio quality and output audio quality? Also the "overall preference" score is unclear - what does this mean, and is this meant to indicate preference over quality 1 vs quality 2?
* What are the various objective scores listed in Table 2?
* The discussion in section 6 is limited and does not provide many points for critical discussion nor any directions for future work - please expand.
* The impact statement does not take into account possible negative implications of such work on the livelihoods of audio engineers and music producers who would normally be the ones carrying out restoration and mastering operations.

**Strengths And Weaknesses:**

Strengths:
* This is possibly the first work attempting to handle multiple common degradations taking place in a music production environment under a single model, which also takes natural language as (optional) input.
* The overall approach is technically correct, using appropriate datasets, degradation methods, and an appropriate generative architecture. Experiments are well setup using common objective metrics plus a listening test for subjective evaluations.

Limitations:
* The paper claims to address both music restoration and mastering. In reality, most of the tasks addressed here would fall under restoration as opposed to mastering. Mastering is a process which does not involve a single snippet or a complete music recording, but also operates at an album level in order to ensure coherence. This is absent from the paper however.
* The discussion in section 6 is limited and does not offer any particularly useful points for critical discussion nor future directions.
* The setup of the listening test is unclear (see comments below).
* The impact statement does not fully list potential social and ethical issues stemming from this work (see comments below).

---

> ### Author Rebuttal · Authors · 2026-03-31
>
> Dear Reviewer,
>
> Thank you for reviewing our paper and your valuable feedback. Please see our reaction to your points below:
>
> - ***As mentioned above, the paper claims to address both music restoration and mastering, however in my view ...***
>
> - ***The discussion in section 6 is limited...***
>
> Thank you for this valuable feedback. Regarding the definition of mastering, we agree it is a broader term that can include more than just single recording modifications, such as album level mastering. The current paper title including the keyword “towards” is meant to suggest this is a start to this new direction and should open the way for follow-up work that can further expand the concept to album-level editing and more.
>
> However, we would also like to highlight that the current narrative focuses on amateur musicians and their needs. Such users are expected to have single-track audio recorded on their phone / cheap microphone with potential artifacts, imbalances, and no option of mixing. Nevertheless, to target both the raised points, we decided to add a whole new limitations section in the appendix and updated the discussion section with the following part:
>
> *“Regarding further limitations, while this work dealt with 19 different degradations, the list is far from exhaustive. The language used to describe the desired change can be further expanded by more musical jargon. Last but not least, this work deals with mastering in a narrow sense, focusing on single track improvement (for amateur musicians), but omitting broader concepts such as album-level editing for cohesion. These areas present avenues for future work. We provide a full list of limitations in Appendix A.10.”*
>
> - ***The paper's reference style is not correct - please use carefully either \cite or \citep ...***
>
> Thank you for pointing this out. We have noticed the incorrect use of \citet in several spots (and \citep in one) in the manuscript, which was likely an overlooked result of a mass edit. We have corrected this. :-)
>
> - ***Was ethics approval needed...***
>
> Ethics approval for the listening tests was a part of the supporting grant, which we can’t currently disclose due to the anonymity period.
>
> - ***The way the listening test was carried out was unclear...***
>
> Thank you for the question. We carried out two listening tests. In the first one, the focus was on controllability and overall improvement over the degraded outputs. In this case, “quality” was focusing on the perceived quality of the audio, i.e., whether it is distorted in any way. The results demonstrated that SonicMaster improved the quality of the audio, eliminating perceivable issues such as clipping, or overly reverberant audio.
>
> The “overall preference” is supposed to focus on listener enjoyment, it does not focus on the audio quality per se, but on the overall experience. For example, if the intended change was to enhance the boominess of the audio, and the model would tend to amplify the bass frequencies too much, the listener might prefer the original input audio over SonicMaster’s output. Or if the equalization of the audio is uneven, does the user prefer the enhanced output provided by SonicMaster over the unbalanced input? The results demonstrated that the SonicMaster esthetically improved audio across the board. We have renamed “overall preference” to “general preference” and updated the text to better describe the meaning of this score as follows:
>
> *“Using a 7-point Likert scale, participants rated text relevance, input audio quality (Quality1), output audio quality (Quality2), output consistency, and general preference (whether input or output sample is preferred in terms of listener enjoyment and satisfaction on the 7-point scale, with 1 meaning full preference of the input sample and 7 of the output sample).”*
>
> The second listening test compared SonicMaster against baselines on specific task subcategories of dereverberation and equalization.
>
> - ***What are the various objective scores listed in Table 2?***
>
> Thank you for pointing this out. Similarly to Table 1, Table 2 shows average absolute error of various metrics, which are introduced in Section 4.1 and further detailed in Appendix A.5. We acknowledge that the caption of Table 2 does not specifically say “average absolute error” unlike its counterpart in Table 1, and will update this for the camera-ready version.
>
> - ***The impact statement does not take into account...***
>
> Thank you for pointing this out. We have updated the impact statement with the following text:
>
> *“Potential risks include increased competition and lower demand in the music production and mastering field. However, we believe that human expertise is still very far from being replaced and is currently invaluable, even more so given the subjective aspect of music and both specialized expertise of professionals and endlessly diverse needs of musicians and producers.”*
>
> We kindly hope our answers are to your satisfaction.
>
> Authors of SonicMaster

---

> > ### Author Rebuttal · Reviewer_zJqJ · 2026-03-31
> >
> > Most of my comments were addressed, however the broader comment made regarding mastering still stands and I would not agree that this can be compensated by the "towards" term in the paper. In my view it would be best if this term was removed entirely from the contributions claimed, but could be stated as future work.
> >
> > The additional information regarding listening tests, the impact statement, and ethics approval is welcome.

---

> > > ### Author Response · Authors · 2026-04-07
> > >
> > > Dear Reviewer,
> > >
> > > We thank you for your response and acceptance of the additional information we provided. As stated before, we acknowledge your concern and agree that the scope of mastering in this work does not cover the broad sense of mastering.
> > >
> > > To complement the changes already made to the discussion and limitations section regarding this, where we newly state this as a current limitation and propose further exploration and improvement in the broader sense of mastering for future work, we further kindly propose to include a clear distinction between what parts of mastering are covered in our work (single-track enhancement) and what the broad term includes in the introduction section.
> > >
> > > Our current proposal is to include the following in introduction:
> > >
> > > >*In this work, mastering refers to targeted perceptual enhancement of degraded music tracks toward professionally polished quality, rather than the full scope of traditional professional mastering. Given that this work is first of its kind, the current focus is only on single-track level adjustments, not on album-level processing for cohesion.*
> > >
> > > We kindly hope this addresses your remaining comment and look forward to your re-evaluation of the paper. Given that this is our last allowed response, we politely ask if you might consider raising your score to reflect these resolved points.
> > >
> > > With Thanks and Regards,
> > >
> > > Authors of SonicMaster

---

### Official Review · Reviewer_GEyH · 2026-03-11

**Soundness:** 2
**Presentation:** 2
**Significance:** 2
**Originality:** 2
**Overall Recommendation:** 3
**Confidence:** 4

**Summary:**

Paper proposes a flow-match model to perform music restoration and mastering against 19 diverse degradations, following natural language prompt/instruction. A text-conditioned music-restoration corpus (audio pairs and text annotations) is constructed to train the model. Experimental results indicate audio quality improvements compared to established baselines (e.g., in Reverb degradation, in Dynamic/Reverb removal, etc.) and degraded audio, while ablation studies suggest a degree of generalization capability in the Piano task.

**Compliance With Llm Reviewing Policy:**

Affirmed.

**Key Questions For Authors:**

Is there stronger baselines can be added into experiments?
Suggest improving the experiment section so that it highlights the main findings more clearly.

**Limitations:**

Yes.

**Strengths And Weaknesses:**

Strength:
Propose one model for music restoration and master, cover diverse (19) degradations across different (5) groups, and provide natural language controllability.

Weakness:
Lack innovation in model architecture, and training strategy. Mainly an application/experiment on existing method.
Full synthetic data may not cover real studio artifacts.

---

> ### Author Rebuttal · Authors · 2026-03-31
>
> Dear Reviewer,
>
> Thank you for reviewing our paper and your feedback. We would kindly like to react to your evaluation of our paper below.
>
> **Concept and contributions of our work:** We would like to highlight this work is a first of its kind application introducing a new task of combined music restoration and mastering and the very first time music mastering is seen as a generative task. The model is designed to be user-friendly by replacing sliders with prompt-based text control, allowing amateur musicians and users without musical knowledge to utilize it. The paper introduces a large dataset of audio with detailed description of parameters used to generate degraded versions of the originals, which allows for a complete reproducibility and enables other tasks in the field, such as parameter prediction, plus text prompts associated with the change. All these contributions were our primary research focus (architectural novelty was not a priority), which is why we submitted this paper to the Application track and feel that it fits there very well with substantial contributions to the field and a big potential for follow-up research.
>
>
> **On the realness and syntheticity of audio data:**
> Obtaining large-scale paired data from real-world music production remains challenging due practical constraints such as copyright restrictions, finance, or simply none availability.
> This is why we designed our data processing pipeline to emulate real conditions, starting with filtering available music data by production quality to ensure professional level of production, and ensuring a variety of genres in our dataset, yielding the clean audio dataset.
>
> Furthermore, in the degradation step, a big portion of the functions used for degrading the clean pieces either comes from the real world, or is well based on real scenarios. For example, we used a set of real recorded room impulse responses, a set of recorded phone microphone transfer functions, or deployed a multiband equalizer with a random number of bands and attenuation selected from a range (8 to 12 bands, -6 to +6 dB in each), which creates a lot of variety in the data, thus accounting for many potential real-life scenarios (see a full list in Appendix A.4, Table 6). Each of the applied functions contains a level of randomness in the parameter selection, except for the stereo function, in which the operation is consistent, but the designed condition to apply this function ensures it is only applied to the more spacious music pieces.
>
>
> **Baselines:** Given the unique nature of SonicMaster being a text-controlled all-in-one music restoration and mastering model, there is no fully equivalent baseline at this time. This is why we were forced to do piece by piece comparisons in different aspects (function categories) to specialized baselines. To the best of our knowledge, we searched for and included suitable baselines, such as the BABE2 model to compare the general audio restoration ability, or Text2FX to compare in terms of text-controlled equalization capabilities. In dereverberation, we present 4 various baselines for comparison. **Overall, we present a total of 15 baselines** to compare our model to (2 variants of Text2FX, Mel2Mel+Diffwave, 2 settings of HPSS, WPE, LTAS-EQ, BEHM-GAN, BABE, BABE-2, DPTNet, UMX, DCUNet, TCN, HDemucs).
>
> Furthermore, we provided a large array of ablation experiments, including architectural scaling, text controllability, inference and training conditioning, and ODE solver experiments.
>
>
> **Regarding the reviewer’s key question:** We touched on the reasoning for baselines in the paragraph above. If you have any specific non-commercial baselines in mind, please feel free to suggest. However, as highlighted, comparisons are never truly fair given the multi-functional ability of SonicMaster and no existing model performing the same task of text-controllable all-in-one music restoration and mastering available for comparison.
>
> We kindly hope our answers are to your satisfaction and look forward to a fruitful discussion.
>
> Authors of SonicMaster

---

> > ### Author Rebuttal · Reviewer_GEyH · 2026-04-01
> >
> > The authors have addressed major review comments. Reviewer notes the use of multiple baselines across different evaluation tasks based on function categories.
> > While this is the first work to create a unified model for music restoration and mastering, similar ideas exist in related fields like audio editing and mastering (e.g. Audit), including similar models (diffusion vs. flow match) and data synthesis approaches, so the novelty should not be overstated.
> > For limitations with synthesized data, more emulations were added for broader coverage, but they do not guarantee real-world artifacts are fully represented; this limitation should be explicitly acknowledged.

---

> > > ### Author Response · Authors · 2026-04-07
> > >
> > > Dear Reviewer GEyH,
> > >
> > > We sincerely thank you for engaging with our rebuttal and for your constructive follow-up feedback. We appreciate the opportunity to properly contextualize our work's novelty and clarify our approach to real-world artifacts.
> > >
> > >
> > > ### 1. Distinction Between Music Mastering and General Audio Editing (AUDIT)
> > >
> > >
> > > We acknowledge and agree that instruction-guided latent diffusion models like AUDIT represent powerful generative methods. However, we would like to respectfully point out the difference between AUDIT and SonicMaster and why the two cannot be properly compared.
> > >
> > >
> > > AUDIT mainly focuses on audio editing tasks, such as adding, dropping, or separating audio. All these tasks represent a change of content present in the audio.
> > >
> > > SonicMaster, on the other hand, focuses on altering the character of the audio present, while retaining all of the original content. SonicMaster deals with polyphonic music and addresses comprehensive multi-artifact music restoration and mastering through a single controllable rectified-flow architecture. Music mastering requires applying intentional tonal and dynamic shaping simultaneously. Bridging dereverberation, declipping, tonal rebalancing, dynamics, and stereo enhancement under prompt guidance without destroying the delicate coherence of a musical track is a distinctly different challenge than localized audio editing.
> > >
> > >
> > > ### 2. Real-World Artifacts and Dataset Limitations
> > >
> > >
> > > We note your acknowledgement of our coverage of real-world emulations (real room impulse responses from the openAIR library, and microphone transfer functions from the POLIPHONE dataset), as described in our initial rebuttal response.
> > >
> > > We would like to further point out another step we took to ensure our model bridges the gap to real-world applications – the historic piano restoration evaluation:
> > >
> > > To further prove our model's capacity to handle authentic, non-synthetic artifacts, we evaluated SonicMaster zero-shot on severely degraded historical solo piano pieces (Section 5.3). Despite lacking domain-specific training for these real-world artifacts, SonicMaster successfully generalized and achieved a **Production Quality (PQ) score of 6.93**. This nearly matches the 7.05 achieved by BABE-2, a state-of-the-art model explicitly specialized for restoring real historical recordings.
> > >
> > > Nevertheless, we do agree that having fully true real-world data (that could include more artifacts) would naturally be beneficial and could improve the model further. We have updated the newly included limitations section with the following statement:
> > > >***Dataset:*** *When creating the SonicMaster dataset, we tried to emulate real-world conditions by: 1) specifically applying real-world room impulse responses from the openAIR library (Howard & Angus, n.d.) and phone microphone transfer functions from the POLIPHONE dataset (Salvi et al., 2025); 2) implementing random ranges for a large number of parameters, covering all degradation functions except for the amplitude and stereo functions, which contained a limited number of options.*
> > >
> > > >*We note that further improvements to the data can be achieved by: a) expanding the dataset to cover more functions, artifacts, genres, instruments, and more; b) collecting a real-world based dataset to better capture and represent any (and all) potential artifacts present in the real world.*
> > >
> > > We hope that these clarifications address your remaining concerns. Given that this is our last allowed response, we politely ask if you might consider raising your score to reflect these resolved points.
> > >
> > > With Thanks and Regards,
> > >
> > > Authors of SonicMaster

---

### Official Review · Reviewer_tirq · 2026-03-12

**Soundness:** 2
**Presentation:** 2
**Significance:** 2
**Originality:** 3
**Overall Recommendation:** 3
**Confidence:** 4

**Summary:**

The authors propose SonicMaster, a generative neural network that can apply mastering and restoration based on natural language text prompts.The authors also release a text-conditioned music restoration dataset.

**Compliance With Llm Reviewing Policy:**

Affirmed.

**Final Justification:**

Fundamentally, this paper outlines how a neural network model can use input text (language) to alter audio towards some desired end. The authors do not meaningfully engage with a relevant body of literature that has collected ecologically valid data of how mixing and mastering engineers use language to describe their practice of altering audio. The claim that this architecture is meant "for beginners" further raises my concerns, as (1) there is no feasibility study of how beginners or experts use language to describe how they want their audio changed; (2) there is no academic/professional consensus on one-to-one mappings of the terms mixing and mastering engineers use to describe how they apply effects to audio towards some goal; and (3) deploying this type of top-down epistemological approach to beginners in the field can discipline these users in language and audio effect usage out of step with reality and artistic practice.

As such I will not be updating my score.

**Key Questions For Authors:**

How can this system account for the broad terminology used by mastering engineers?

Please provide the full language for the listening test performed in this study.

**Limitations:**

The authors should discuss the limitations of the language they have used in their dataset.

**Strengths And Weaknesses:**

Soundness:

The application of neural network machinery in this paper is all quite sound and meaningfully engages with state-of-the-art techniques.

My primary concern with this paper is the reductive approach it takes to the language that mastering engineers use in their practice, as well as what they are expected to do at the mastering stage of music production. Mastering and restoration are quite different fields, and many of the “asks” provided by the prompts are mixing asks. I would encourage the authors to explore ethnographic work in the field of music production (published in the Audio Engineering Society and Society for Music Production Research, see the work of Amandine Pras) to see how researchers study the use of language across diverse backgrounds in these fields.

In addition, the evaluation of the proposed system included no comparisons with commercially available tools for automatic mastering. Though I do not believe these tools are text-guided at the moment, their inclusion in a perceptual evaluation, as a “textless guidance benchmark,” would strengthen claims regarding the utility of text descriptions for improved mastering and restoration.


Presentation:

The technical presentations of neural network techniques in this paper are sound.

The introduction of this paper reads more like a paper on automatic mixing than automatic mastering. For example, much of the research Wilson and Mourgela have published deal with multitrack mixing, where fixes can be applied to raw tracks prior to mastering.

The perceptual evaluation is poorly described. For example, there is no clear indication for what “overall preference” refers to.


Significance:

There are several automatic mastering solutions available for purchase, but none that I am aware of that can be guided with unstructured text prompts. This work’s significance could be clearly demonstrated via a user feasibility study, which could help test the universality of the language used to generate the dataset in this paper.


Originality:

This is an original approach to the best of my knowledge.

---

> ### Author Rebuttal · Authors · 2026-03-31
>
> Dear Reviewer,
>
> We thank you for taking the time to review our paper and for your valuable feedback, particularly for recognising our neural network machinery is highly sound, meaningfully engages with state-of-the-art techniques, and presents an original approach. Please find our responses to your points below:
>
>
> **Distinction Between Mixing, Mastering, and Restoration:**
> We agree that these are distinctly different fields in professional pipelines. However, SonicMaster is specifically designed for non-professional-setting music where the user only has access to a single rendered audio file, not the raw multitrack stems. Because in such settings, creators cannot go back and fix the raw tracks in the mix, they must apply these "mixing tasks" (like declipping) directly to the rendered audio file. This forces a convergence of restoration and mastering into a single process. We will explicitly clarify this distinction, emphasizing that SonicMaster addresses the reality of non-professional production where mixing fixes must retroactively be applied during the mastering stage.
>
>
> **Evaluation:**
> We apologize for the lack of clarity regarding the overall preference metric in our listening test, we have now renamed it to “general preference” and updated the paper with the following clarification:
>
> *“Using a 7-point Likert scale, participants rated text relevance, input audio quality (Quality1), output audio quality (Quality2), output consistency, and general preference (whether input or output sample is preferred in terms of listener enjoyment and satisfaction on the 7-point scale, with 1 meaning full preference of the input sample and 7 of the output sample).”*
>
> Regarding the comparison with commercial products, we decided not to compare to them given their black-box and non-open-source nature, as it would not be a fair comparison (they do not offer text control) and would not provide insightful knowledge (as we do not know any implementation details about them).
>
> ### **Key questions:**
> ***1) How can this system account for the broad terminology used by mastering engineers?***
>
> To account for the broad terminology used by mastering engineers, SonicMaster leverages the FLAN-T5 Large language model to encode text instructions into a semantic latent space. This allows the model to generalize beyond the specific phrasing used in the dataset, mapping diverse user queries to the appropriate restorative transformations. While our dataset utilizes 8–10 prompt variations per degradation type (Exhaustive details in Table 11, appendix), we agree mastering engineers use nuanced vocabulary. We added this important point in our manuscript and included a whole new detailed section on limitations (touching on all the limitations, including latent space, language, functionality, design choices, and more) in the appendix. We will happily cite ethnographic research in music production, such as the work of Amandine Pras, to better contextualize how language is used across diverse audio engineering backgrounds.
>
>
> ***2) Please provide the full language for the listening test performed in this study.***
>
> To improve clarity in the perceptual evaluation, we now provide the full language of the listening test task in the Appendix.
> Below, we kindly copy the text example for your viewing:
>
> **Briefing text:**
>
> >Hey! Thank you for participating in this listening test. :-)
>
> >After the initial survey questions about your background, you will encounter some 40 questions with audio samples (which should take you some 20-30 minutes). Each question will have 2 samples that form a pair. The first (top) sample represents the input, the second (bottom) sample represents the output after applying a certain transformation described by the text prompt, e.g., "Make this audio brighter" should make the second sample sound brighter compared to the first sample; "Remove the echo!" should make the second audio have less of echoey reverb present than the first audio.
>
> >Please use headphones! You do not need to listen to the whole audio samples, just listen enough to make a conscious judgement.
>
>
> **Example of a description provided with each audio sample pair:**
>
> >Instructions:
> There are two audio samples above. The second one is a modified version of the first one. This modification was done according to the Prompt. Please listen to both the audio clips, read the Prompt, and answer the questions below.
>
>
> >Prompt: Lift the treble for a more open tone.
>
>
> >1) To what extent does the second audio sample represent the intended change given by the Prompt?
> 2) How good is the audio quality of the first sample?
> 3) How good is the audio quality of the second sample?
> 4) How fluent and coherent is the second sample?
> 5) Do you prefer the second sample over the first sample?
>
>
> We hope our answers are to your satisfaction and look forward to a fruitful discussion!
>
> Authors of SonicMaster

---

> > ### Author Rebuttal · Reviewer_tirq · 2026-04-03
> >
> > (2) Thank you for including the text of your perceptual evaluation.
> >
> > (1) While I appreciate the authors' technical description of their language embedding procedure via FLAN-T5, my primary critique regarding the usage of language in mastering has not been addressed.
> >
> > PhD theses have been devoted to examining the use of language and semantic descriptions in music production and multitrack mixing, see Alex Wilson's 2017 thesis "Evaluation and Modelling of Perceived Audio Quality in Popular Music, towards Intelligent Music Production" and Andrew Charles Parker's 2022 thesis "Towards a Perceptual Model of Clarity in Music Mixes." Additionally, datasets like Mark Cartwright's "Social-EQ" and Brecht DeMan's "Mix Evaluation Dataset" attempt to collate ecologically valid semantic descriptors of music production and audio effect applications from professional and amateurs alike.
> >
> > Given this widespread field of inquiry that generally points to ambiguity in the usage of the terms the authors use to train their model, I question what semantic understanding of the term "dark" the FLAN-T5 model would have prior to any training/finetuning, how the representation of that term changes over the course of the proposed training regime, and finally how that representation aligns with practitioner's understanding of the term. The perceptual evaluation run by the authors is a step in the right direction, though I worry about priming participants' expectations with the given text description. The agreement metric captured by the authors is appropriately collected, but I would expect that if they were prompted to describe how the audio had been altered from input to output (in the absence of a text description), participants responses would vary quite a bit and deviate from the prompts the authors used.
> >
> > These types of questions could begin to be answered with user feasibility studies of the proposed systems, or by engaging with some of the datasets I have mentioned above.
> >
> > As such, I will not be revising my score at this time.

---

> > > ### Author Response · Authors · 2026-04-07
> > >
> > > Dear Reviewer,
> > >
> > > Thank you for your response. We acknowledge your concerns about the language semantics and agree that meanings can vary with user groups as well as on individual level.
> > >
> > > SonicMaster is focused on non-expert users. As such, a thorough study on the language semantics has not been the focus of our current work. We designed our prompts with the knowledge of our table of experts and musicians to try to accommodate the non-expert user group, including people with no musical knowledge (e.g., “remove the church effect” prompt). Our results and ablation study on text controllability demonstrate that the model listens to the proposed prompts and performs the desired operations to a satisfying level.
> > >
> > > Given that a thorough study on the language semantics was out of scope for this work, we would respectfully like to summarize the scope and focus of the work, which we feel is already very substantial.
> > >
> > > **Main focus of the work:**
> > >
> > > 1) To showcase a combination of music restoration and mastering in one model is a feasible solution and can be achieved through a **generative paradigm.**
> > > 2) To showcase free flow text control can steer such a model in the desired task.
> > > 3) It aims to enable amateur musicians to overcome a barrier of music knowledge and expensive tech by providing them with a model that is easy to control (no sliders and difficult terms).
> > > 4) To open this new direction of research and bring valuable knowledge to the community.
> > >
> > > **Scope of work:**
> > >
> > > 5) We evaluated SonicMaster against a total of **15 baselines.**
> > > 6) Our ablation study further focused on architecture scaling, conditioning during training and inference, ODE solvers, and text control.
> > > 7) We performed 2 listening studies.
> > > 8) We introduce a new dataset which contains very detailed metadata, making it fully reproducible (we will provide the full dataset in the camera-ready version) and transparent. This dataset can be utilised for other tasks, such as parameter prediction.
> > > 9) We provide an overwhelmingly large appendix, which gives very detailed insights into all the degradation functions used, the designed prompts, and the evaluation metrics.
> > > 10) All our data, models, codes, and demos are open source, allowing for a high degree of reproducibility.
> > >
> > > **About the current semantics:**
> > >
> > > The semantic understanding of the term “dark” is learnt through the training process, where the input prompts are mapped to the relevant operations given the dataset triplets of (input audio, output audio, instructing prompt), as is the case with the rest of the proposed terms. This mapping also results in a semantic alignment of the different prompts used for the same function. In practical terms, this means that the language semantics of our model are determined by the design of our prompts and are unique to it. It does not follow any “standard” so to say. Given that SonicMaster is not a commercial product, but a piece of research work with focuses other than language semantics, we respectfully find the suggested language feasibility study rather a potential future direction than a current weakness.
> > >
> > > We would further like to pinpoint that the concept of instructions in SonicMaster is to provide a direction of change – the prompt does not describe the desired end state, such as “dark”, but it points to a direction of change, i.e., “darker” means to shift the character of the input audio to contain less high frequency content, but it does not mean the result is “dark” if the input audio already contained too much of high frequency content.
> > >
> > > In essence, the inference pipeline is: input audio -> instruction describing the direction of change -> output audio moved in this desired direction.
> > > To shortly demonstrate the learnt meanings of the terms, we invite the reviewer to listen to the samples provided on our anonymous website:
> > > https://msonic793.github.io/SonicMaster/
> > >
> > > **Conclusion:**
> > >
> > > Finally, to conclude and further address your comments, we will gladly incorporate the valuable points on language semantics and semantic ambiguity by dedicating a few lines to the language semantics in the introduction section, and further clearly include this in the new limitations section and propose it as an important challenge and avenue for future work.
> > >
> > > We hope our response helped clarify the matter. Given that this is our last allowed response, we would like to ask the reviewer whether they might consider updating the final score.
> > >
> > > With Thanks and Regards,
> > >
> > > Authors of SonicMaster

---

### Decision · Program_Chairs · 2026-04-30

**Decision:**

Accept (regular)

**Comment:**

The authors propose SonicMaster, a unified flow-matching generative model designed for text-controllable music restoration and track-level enhancement. To train the model, the authors introduce a new dataset of paired degraded and high-quality tracks, simulating 19 degradation functions across 5 enhancement groups.

The paper presents a commendable engineering effort and a valuable dataset for text-conditioned audio restoration. However, it struggles with domain-specific terminology and semantic ambiguity. The authors successfully addressed several minor concerns (clarifying listening tests, updating impact statements, and adding limitations), but the fundamental disagreement regarding the epistemological approach to audio engineering language and the scope of the term "mastering" remain unresolved.

Despite these limitations, the unified model and dataset represent a solid contribution to the Applications track, opening a new direction for non-professional audio enhancement.